# Genomic and transcriptomic changes complement each other in the pathogenesis of sporadic Burkitt lymphoma

Cristina López ● et al.[#]

Burkitt lymphoma (BL) is the most common B-cell lymphoma in children. Within the International Cancer Genome Consortium (ICGC), we performed whole genome and transcriptome sequencing of 39 sporadic BL. Here, we unravel interaction of structural, mutational, and transcriptional changes, which contribute to *MYC* oncogene dysregulation together with the pathognomonic IG-*MYC* translocation. Moreover, by mapping IGH translocation breakpoints, we provide evidence that the precursor of at least a subset of BL is a B-cell poised to express IGHA. We describe the landscape of mutations, structural variants, and mutational processes, and identified a series of driver genes in the pathogenesis of BL, which can be targeted by various mechanisms, including IG-non *MYC* translocations, germline and somatic mutations, fusion transcripts, and alternative splicing.

Burkitt lymphoma (BL), including its leukemic counterpart Burkitt leukemia (B-AL), is a highly aggressive lymphoid neoplasm supposed to derive from germinal center B (GCB) cells[1]. It is the most common B-cell lymphoma in children but also occurs in adults[2,3]. The monomorphic medium-sized tumor cells express membrane IgM with light-chain restriction and typical B-cell antigens, including CD19, CD20, and BCL6[1].

Three epidemiologic variants of BL are distinguished: endemic, sporadic, and immunodeficiency associated[1]. Endemic BL is the predominant form around the malaria belt in Africa. It frequently involves jaw and facial bone, and is closely linked to Epstein-Barr virus (EBV) infection[4]. Immunodeficiency-associated BL is related with inborn or acquired types of immune deficiency. It presents frequently as nodal disease and EBV infection is documented in 25–40% of patients[5,6]. Sporadic BL (sBL), is the most common form of BL outside the malaria belt. It presents mostly as an abdominal disease particularly in the ileocecal region or in lymph nodes. Leukemic infiltration accompanies tumorous presentation in a subset of sBL. Nevertheless, the term B-AL is restricted to cases with leukemic infiltration exceeding 25% of cells in the bone marrow[1]. EBV is found in 10 to 20% of sBL.

The genetic hallmark of all three BL types is the IG-MYC translocation involving the MYC oncogene and mostly the immunoglobulin heavy chain (IGH) locus, or more rarely, one of the immunoglobulin (IG) light-chain loci[6–8]. Subsequent dysregulation of MYC expression has been shown to be driven by the respective IG enhancer[9].

Studies in mice and humans have shown that deregulation of MYC alone is not sufficient to drive BL lymphomagenesis[10–12]. Cytogenetically, BL shows a low genomic complexity, with the IG-MYC translocation being the sole abnormality in around 40% of cases[7,13,14]. Nevertheless, recent sequencing studies identified recurrent somatic mutations in ID3, TCF3, CCND3, and SMARCA4 in both, sporadic and endemic BL[15–18]. However, these studies mostly applied exome, transcriptome, or targeted sequencing strategies, neither taking into account the non-coding genome nor systematically integrating the various layers of nucleic-acid encoded information. Investigating a limited set of 13 BL for epigenetic changes, we recently provided initial evidence for a tight connection between somatic mutation, DNA methylation, and transcriptional control in BL[19]. Here, in the framework of the International Cancer Genome Consortium (ICGC), we performed a comprehensive analysis of whole genome (WGS) and transcriptome (RNA-seq) sequencing data and extended our prior series to 39 sBL in children treated in prospective clinical trials. By integration of the different datasets we provide insights into the complex genomic and transcriptomic changes underlying MYC dysregulation, the potential cell of origin of sBL, and the complementarity of mutational mechanisms deregulating key pathways in BL.

## Results

**Study cohort.** Using the inclusion criteria outlined in Supplementary Fig. 1a and the Method section, we included a total of 39 IG-MYC single hit (i.e., without a further translocation event affecting BCL2 and/or BCL6), EBV/HIV negative sBL from children (≤ 18 years) with a median age at diagnosis of 8 years (range 2–18) into this study. The male:female ratio in our cohort is 6.5:1. All patients were registered in one of the clinical trials of the Mature B-cell-non-Hodgkin-Lymphoma, Berlin-Frankfurt-Münster (NHL-BFM) study group. At last follow-up (median time: 67 months), 37 patients were alive without evidence of disease. The clinical parameters of the 39 patients were comparable to the overall B-NHL population treated according to the Berlin-Frankfurt-Münster (BFM) multicenter protocols[20] (Supplementary Table 1).

Considering the arbitrariness of the 25% cut-off for distinguishing B-AL from BL according to the World health Organization (WHO) classification[1], and given that the percentage of bone marrow infiltration can vary according to site of aspirate within a patient, we refrained from differentiating B-AL from BL. Instead, we classified the cases according to the site where the lymphoma cells were derived from for sequencing. Thus, we distinguished solid manifestation of BL, e.g., in lymph nodes or solid masses (solBL, $n = 27$), from non-solid BL forms; the latter including leukemic manifestations in bone marrow (leukBL, $n = 9$), and fluidic manifestation in pleura (pleuraBL, $n = 3$) (Supplementary Fig. 1a and Supplementary Data 1).

**Molecular classification of BL by transcriptional profiling.** To confirm the diagnosis of BL on the molecular level, we adapted the previously published molecular BL (mBL) gene expression index[21] to RNA-seq data available for 23 of the 39 patients (16 BL cases without suitable RNA). For RNA-seq, a median of 138 million reads (range 95–178 million) were obtained per sample, of which 99.76% mapped to the human genome (Supplementary Data 2). All 20 solBLs with RNA-seq data were classified as mBL. However, two of three leukBL were classified as non-mBL and all three leukBL showed an expression pattern more typical for non-mBL for the majority of genes in the mBL classifier (Supplementary Fig. 1b). This is in line with our previous findings using the array-based classifier and most likely due to the fact that the "mBL index" was developed to classify BL in solid masses like lymph nodes[21]. Given these findings, we focused only on the solBL cases for all further expression analyses using the RNA-seq data.

**Frequency of genomic alterations determined by WGS.** We performed WGS of tumor and matched normal tissues of all 39 patients with a mean coverage of 38x and 40x, respectively, and detected overall 98,914 somatic single-nucleotide variants (SNVs) cohort-wide, corresponding to a median of 2304 somatic SNVs per sample (range 1178–5279) (Supplementary Fig. 1c). A total of 1238 (1.25%) were predicted to affect protein function, with a median of 29 per sample (range 16–61). A total of 8214 somatic small insertions/deletions (indels) were detected, corresponding to a median of 218 somatic indels per sample (range 75–601), of which 98 were detected in coding regions, with a median of three coding indels per case (range 1–8). We investigated the aberrations across the BL groups, and observed a significantly lower number of total coding and total non-coding events in leukBL as compared to solBL (24 vs 35, $p = 0.008$, and 1901 vs 2568, $p = 0.002$, respectively, Kruskal–Wallis Test), and as compared to pleuraBL (24 vs 32, $p = 0.050$, and 1901 vs 2954, $p = 0.021$, Kruskal–Wallis Test) (Supplementary Fig. 2). A median of 24 high-confidence structural variants (SVs) were found per case (range 7–289) including a median of eight deletions (range 2–274), two duplications (range 0–12), four inversions (range 0–89), and three translocations (range 0–14, please note that some IG-MYC translocations were not considered as they had scores <3, i.e., below the high-confidence threshold, please see Methods section). Given the low number of SVs as compared to other germinal center derived B-cell lymphomas (GCB-lymphomas), we explored the role of telomere content and function in BL. Though TERT transcripts are highly elevated in BL compared to normal GCB cells ($p < 0.001$) the observed increase of telomere content between tumor cells and the normal GCB cells controls was not significant (Supplementary Fig. 3).

**Characterization of the IG-*MYC* translocation**. We detected the pathognomonic IG-*MYC* translocation, in all samples by fluorescent in situ hybridization (FISH) and/or WGS (Supplementary Data 1). Case 4177434 harbored a complex IGH-*MYC*-*ASIC2* translocation, which could only be resolved by the combination of cytogenetics and sequencing techniques (Supplementary Fig. 4). An IGH-*MYC* juxtaposition was detected in 33 BL, an IGK-*MYC* in one case and IGL-*MYC* in five cases (Supplementary Data 1). Different processes remodeling the IG loci have been shown to be involved in the formation of oncogenic IG translocations in B-cell lymphomas, namely aberrant VDJ recombination, class switch recombination (CSR) and somatic hypermutation (SHM)[9]. To elucidate the mechanisms contributing to the generation of the IG-*MYC* translocations in the analyzed BL, we investigated the IG breakpoints in detail. We could map the exact breakpoint location at the IGH locus for 32 out of 33 cases with IGH-*MYC* translocations using WGS. We observed a translocation of *MYC* to the IGHM switch region in 12 cases (37.5%), to one of the IGHA switch regions (IGHA1 = 3 and IGHA2 = 5 cases) in 8 cases (25%), to IGHG switch regions (IGHG1 = 3 and IGHG3 = 3 cases) in 6 cases (18.7%), and to the VDJ region in 6 cases (18.7%). Of the variant IG-*MYC* translocations, five involved IGLJ and IGLV regions, and one the IGKJ5-IGKC intervening region. Next, we performed a detailed analysis of the junctional sequences for the potential mechanism resulting in the translocation. This was feasible in 33 cases. Three cases (4127766, 4158769, and 4189998) were excluded because the sequences were not informative, two (4193278 and 4103570) because it was not possible to determine the mechanism causing the translocation (Supplementary Data 3), and finally case 4177434 with the complex IGH-*MYC*-*ASIC2* rearrangement has not been considered for this analysis. Taking into account all three IG loci, aberrant CSR was the predominant mechanism leading to IG-*MYC* translocation (24/33 informative cases, 73%), followed by SHM involved in 9/33 informative cases (27%). Of the latter, 4 had the IGL loci as *MYC* partner. Thus, SHM was the underlying mechanisms resulting in the translocation in 5/28 BL with IGH-*MYC* and 4/5 BL with IGL-*MYC* juxtaposition, respectively. Remarkably, a misled VDJ recombination as cause of IG-*MYC* juxtaposition was not identified in any case. We also explored the location of the breakpoints in the IGH locus within solBL, leukBL, and pleuraBL separately. The leukBL cases displayed only breakpoints in the switch IGHM or VDJ regions. In contrast, solBL and pleuraBL showed breakpoints distributed over the whole IGH locus (Supplementary Fig. 5a). For formal testing of these different distributions, we classified the breakpoints into two groups: centromeric of IGHM (i.e., switch regions upstream of constant regions involved in later phases of the immune reaction) and within the IGHM region/or telomeric of IGHM (including VDJ) (Supplementary Fig. 5a). The location of breakpoints in the IGH locus differed significantly between leukBL, solBL, and pleuraBL ($p = 0.031$, Pearson Chi-Square test); the leukBL showed the breakpoints within or telomeric of the IGHM region, whereas solBL and pleuraBL displayed breakpoints distributed over the whole IGH locus. Finally, we extended the study of the IGH breakpoints to adult GCB-lymphomas other than BL with break in IGH locus. We included follicular lymphoma (FL) ($n = 84$), diffuse large B-cell lymphoma (DLBCL)($n = 75$) and FL-DLBCL ($n = 17$), as well as double hit (DH) lymphomas ($n = 2$) with mBL signature and B-cell lymphoma not otherwise specified (B-NOS, $n = 1$) from the ICGC MMML-Seq project (unpublished data). In this extended cohort we observed a significant skewing of the IGH breakpoints in BL to the IGHA switch regions (Supplementary Fig. 1a). Indeed, oncogene-IG-translocations with breakpoints in the IGHA switch region were only observed in cases with *MYC* as partner and were almost

exclusive to BL ($p < 0.0001$) in the complete ICGC MMML-Seq population (Supplementary Fig. 5b-c). The only exception was a DH lymphoma with mBL signature presenting in the tonsil of a 75-year-old patient that displayed an IGH-*MYC* fusion with a breakpoint in the IGHA1 switch region (case 4182605). We cannot exclude that this case is indeed also a bona fide BL despite the DH. Given that CSR is supposed to require transcription[22], we investigated the IGHA RNA expression in relation to the breakpoint location in the BL cohort. We observed a significantly higher expression of IGHA transcripts in cases with IGH-*MYC* translocation breakpoints mapping to the IGHA region ($p = 0.0012$, Wilcoxon test).

Next, we analyzed the breakpoints of the IGH-*MYC* fusions on chromosome 8 (*MYC*) in more depth. These have previously been classified in three categories according to the position of the chromosomal breakpoints relative to the *MYC* gene[23]. We observed class I breakpoints affecting the first exon or intron of *MYC* in 17 of 32 cases (53%, case 4177434 with IGH-*MYC*-*ASIC2* rearrangement not considered for this analysis), and class II breakpoints located immediately upstream of *MYC* in 14 of 32 (44%) (Supplementary Data 3). No class III breakpoint far upstream of the 5′ end of the gene was detected. However, unexpectedly one case with an IGH-*MYC* translocation, had a breakpoint downstream of *MYC*. In contrast to the classic *MYC* translocation involving IGH, variant *MYC* translocations to IGK and IGL ($n = 6$ cases), showed breakpoints downstream of *MYC* in line with previous studies[23,24].

To gain deeper mechanistic insight into the BL hallmark event we studied the effects of the different IG-*MYC* translocations on *MYC* gene expression (Fig. 1b). High RNA expression of MYC was observed in BL as compared to normal GCB cells (fold change 11.78, $p < 0.0001$, Wilcoxon test, Fig. 1c). MYC transcript levels were similar in cases with IGH-*MYC* and with the variant IGK/IGL-*MYC* translocations. Furthermore, we investigated allele-specific MYC transcript expression using somatic sequence variants in the transcribed regions as surrogate marker of the translocated *MYC* allele (IGH-*MYC*). We observed that the vast majority of MYC transcripts originate from the translocated allele (Fig. 1d). Remarkably, we identified two different forms of MYC transcripts that were associated with the location of the translocation breakpoints. Canonical MYC transcripts mainly occurred in cases with class II breakpoints, whereas expression of an alternative MYC transcript using an alternative transcription start site within the canonical first intron was linked to class I breakpoints (Fig. 1b). The mRNA produced from the alternative transcript contains 486 nucleotides from this intronic sequence (chr8: 128750008–128750493 bp, hg19), which are not included in the canonical transcript. Nevertheless, this alternative transcript is predicted to encode the same MYC protein as the canonical transcript, because the start codon predominantly used for MYC protein expression in BL is located in exon 2 of the gene (isoform 1: P-01106–1)[25]. Moreover, the additional 486 nucleotides do not contain an open reading frame continuing into the coding sequence in exon 2. Surprisingly, we observed one case (4125240) exhibiting expression of the first exon of MYC, indicative of the canonical transcript, but also the intronic sequence, which is part of the alternative transcript. There were hardly any sequencing reads spanning the splice junction between the canonical MYC exons 1 and 2 (Supplementary Fig. 6). Combining data from WGS, RNA-seq, and FISH we were able to reconstruct a complex genomic event in this case, which separated exons 2 and 3 (alternative transcript) from exon 1 of *MYC* and brought both parts under the influence of different IGH enhancer elements (Supplementary Fig. 6a).

Previous studies have identified BCL6 binding sites (BCL6BS) in the promoter region of the *MYC* gene, located within 2 kb

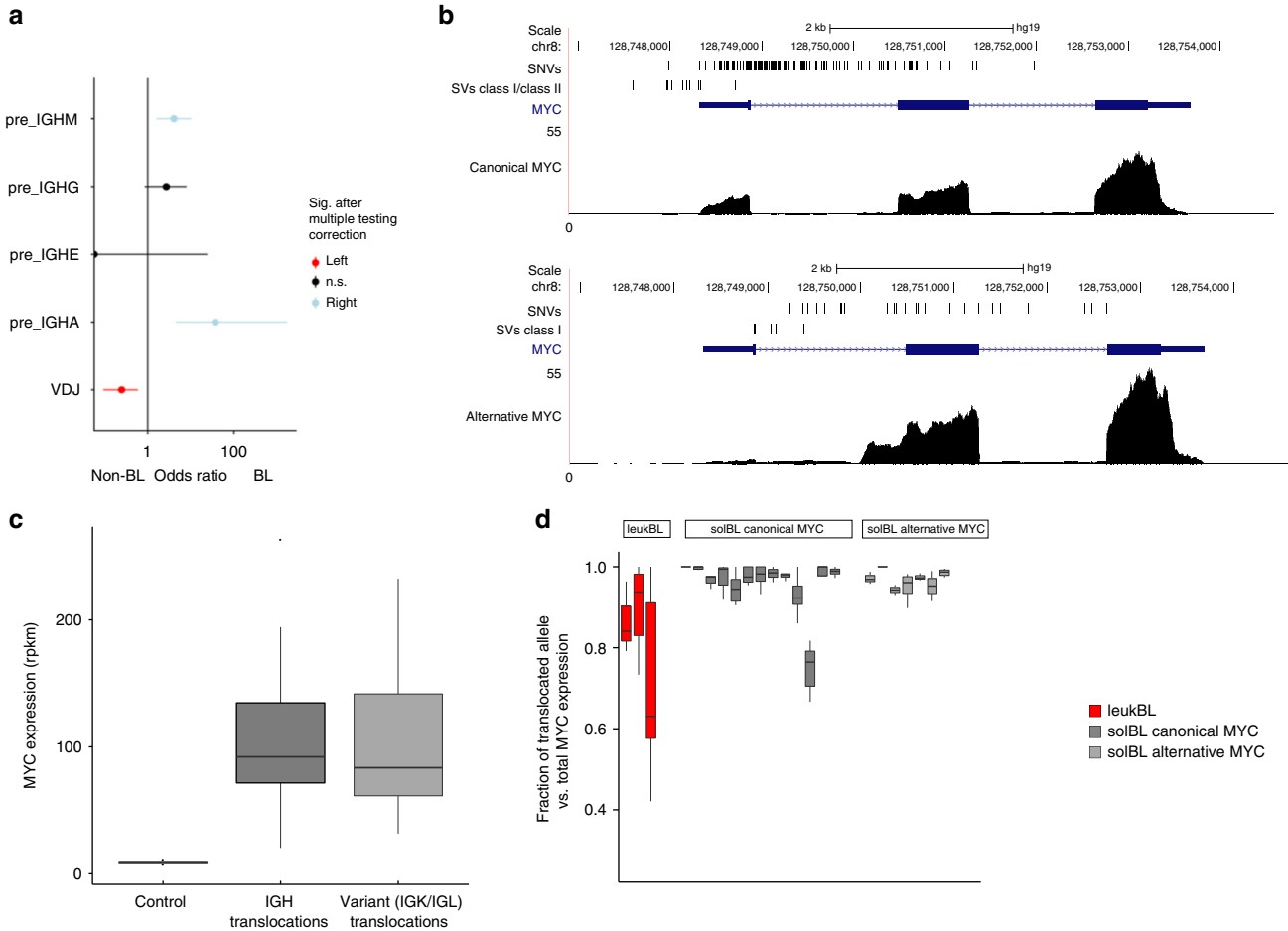

**Fig. 1** Integrative analysis of the IG-*MYC* translocation. **a** Forest plot comparing breakpoint occurrences in the switch region of the IGH locus between BL cases (39 BL studied herein and additional one adult BL) and 179 non-BL GCB-lymphomas, showing a statistically significant enrichment of breakpoints in the switch IGHA regions in BL (Fisher test). **b** Distribution of somatic SNVs and breakpoints (Class I and II) in the *MYC* locus in cases with IGH-*MYC* translocations correlated with the expression pattern of the MYC transcript (upper panel: canonical transcript; lower panel: alternative transcript). Cases with breakpoints downstream of *MYC* and/or without available RNA-Seq data are not shown. The genomic coordinates (hg19) on chromosomal region 8q24 are given as *x* axis on the top. Blue arrows in *MYC* gene indicate the transcriptional orientation on the forward strand. The scale of the *y* axis indicates the mean expression of the respective MYC transcript across the cases using a maximum scale of 55 (normalized value of the expression). Note that due to the resolution not all neighboring mutations can be clearly distinguished. **c** Mean expression of MYC transcripts in normal germinal center B-cells from 5 donors (control) (black) and BL cases with the typical IGH-*MYC* translocation (*n* = 18) (dark gray) or its variants, IGK- or IGL-*MYC* (*n* = 5) (light gray). **d** A predominant expression of the translocated allele, irrespectively of canonical or alternative transcript, is determined by allele-specific expression analysis of MYC by comparing the fraction of translocated allele vs the total MYC allele expression in each group of the BL cohort. The box plots give the range of the fractions (per somatic SNV) of translocated allele vs total MYC expression per case

upstream of the canonical MYC transcription start site (TSS). BCL6 binding represses MYC transcription in GCB cells[26]. As BL cells express BCL6 and, thus, should suppress MYC expression, we investigated, if the BCL6BS is affected by the translocations using the genomic coordinates previously described (chr8: 128746338-128748338 bp, hg19)[26,27]. Indeed, in all 31 BL cases with IGH-*MYC* translocation and class I/II breakpoints the suppressive BCL6 element is either translocated away from the *MYC* gene (*n* = 15) or directly affected (*n* = 16) by the breakpoint.

Next, we explored the presence of IGH-*MYC* fusion transcripts. Fusion reads from the der(14) chromosome were observed in the RNA-seq data in 9 of 18 evaluable cases (50%). Remarkably, more than one type of fusion transcript (with maximum of 4) was detected from both derivative chromosomes in 5 of 9 cases as consequence of the IGH-*MYC* translocation. We could determine the orientation of those fusion transcripts in five cases and interestingly all of them appeared to be an antisense

transcript. IGH and MYC antisense transcripts have been previously described and linked to IGH switch regions and the SHM and CSR events generated by the enzyme activation-induced cytidine deaminase (AID)[28–30]. Thus, the identified antisense fusion transcription might be attributed to the generation of the translocation breakpoints by this enzyme. In addition, we discovered in four cases fusion transcripts between IGH sequences on der(8) and the lncRNA *CASC11*. This lncRNA is located ~2 kb upstream of *MYC* and transcribed in the same orientation as the IGH locus on der(8) but in opposite direction to *MYC*. As expected, all cases with IGH-*CASC11* fusion transcripts exhibited breakpoints upstream of *MYC* (class II).

Overall, we found that MYC transcription is mainly driven from the translocated allele, on which the BCL6BS suppressive element is either translocated away or disrupted, and that class I breakpoints lead to expression of an alternative MYC transcript. Moreover, associated with t(8;14) we observed IGH-*MYC* fusion transcripts in some of the cases. Furthermore, we detected IGH-

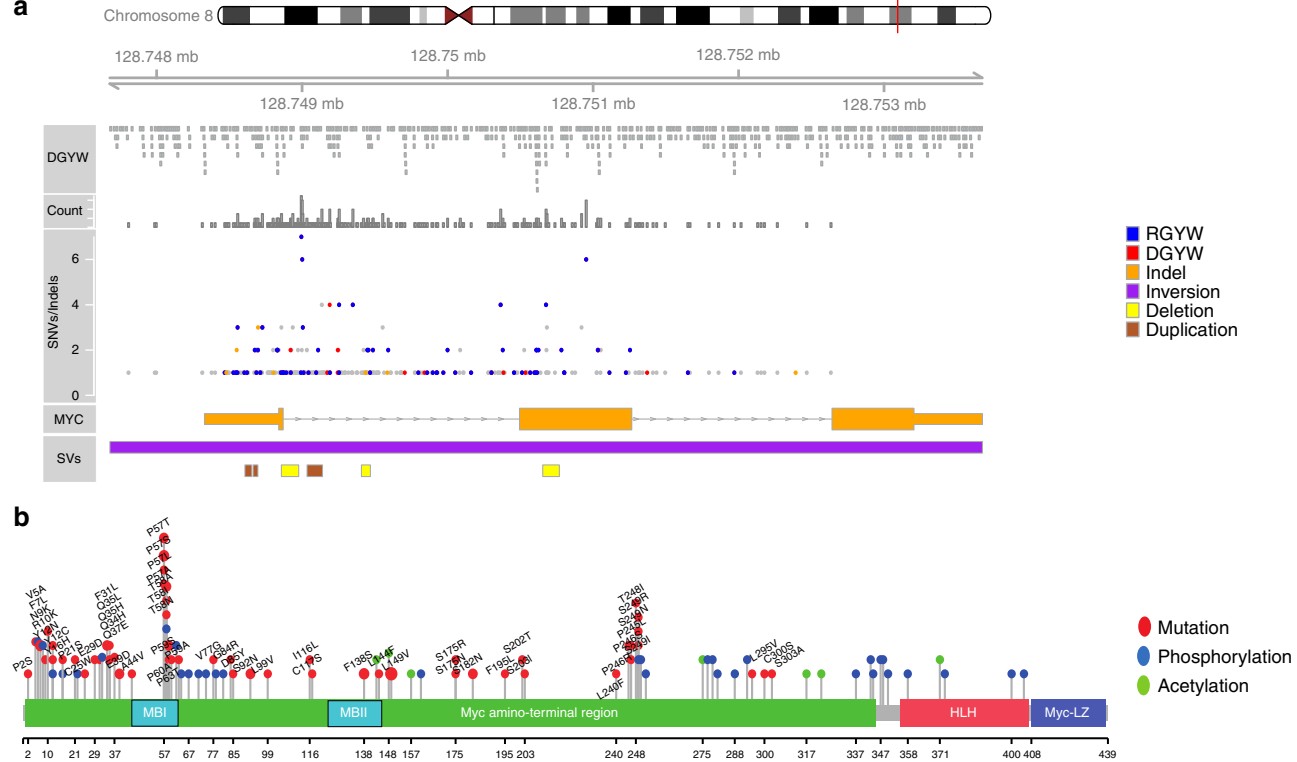

**Fig. 2** Mutational analysis on MYC gene and protein. **a** Distribution of RGYW/DGYW motifs in the genomic region adjacent to and inside the *MYC* gene (upper panel), and the dispersion of somatic SNVs, indels and SVs in *MYC*. Mutations which affect the RGYW/DGYW motifs, are indicated by blue and red dots; gray dots show mutations outside the motifs. **b** Distribution of *MYC* mutations (red lollipops) across the MYC protein sequence (Uniprot: P-01106-1), and the location of the post translational modification (PTM) sites (blue lollipops: phosphorylation sites; green lollipops: acetylation sites). Note that the isoform 2 of the MYC protein (Uniprot: P-01106-2) harbors 15 N-terminal amino acids which are not present in the canonical isoform 1

*CASC11* fusion transcripts, which have not been previously described.

**Analysis of *MYC* SNVs and indels**. Besides genomic translocation, dysregulation of *MYC* can also be caused by SNVs and indels[15,17]. Occurrence of multiple nonsynonymous mutations in the coding sequence of the *MYC* gene have been described in approximately 40–70% of BLs[15,17]. These SNVs cluster in the MYC transactivation domain with hotspots in the Myc box I (MBI) motif (44–63aa)[15,17]. In our cohort, we identified a total of 370 coding and non-coding SNVs in the *MYC* locus (ENST00000377970.2, Fig. 1b). Of these, 184 occurred in cases expressing the canonical and 30 in cases expressing the alternative MYC transcript; the remaining 156 SNVs were in cases where the transcript could not be classified, due to the unavailability of RNA. Of 184 SNVs in cases expressing the canonical transcript, 2 mutations (1%) were located upstream of the transcription start (16 and 343 bp upstream), 31 (17%) were located in the 5′UTR, 23 (12%) in coding regions (8 in exon 1 and 15 in exon 2), and 128 (70%) in intronic regions (125 in intron 1, and 3 in intron 2) (Supplementary Data 4, Figs. 1b and 2a). Of the 30 SNVs in cases expressing the alternative transcript (chr8: 128750008–128750493 bp, hg19), 1 (3.3%) affected the exon 1 of the canonical transcript, 8 (26.7%) intron 1, and 6 (20%) intron 2. These, did not affect the overexpressed MYC transcript. A total of 13 (43%) fell into exon 2 with 5 of them affecting the part of exon 2 exclusive to the alternative transcript (Supplementary Data 4). We also detected 7 small indels in 7 patients, and 7 SVs (3 duplications, 3 deletions, and one inversion) involving *MYC* in five BL; three of them were located in the coding region. In total,

*MYC* aberrations affecting the exonic coding or splice regions were observed in 77% (30/39) of the analyzed BL, including 60 SNVs and 3 SVs. A total of 10 exonic positions were recurrently mutated with 8 of them showing exactly the same base change. The SNVs and SVs clustered in the transactivation domain, affecting the conserved boxes MBI and II (11.1%, 7/63 events) (Fig. 2b). However, a mutational hotspot (6/39 cases; 15%) in our cohort was outside of MBI at position 149 (p.Leu149Val) in the protein sequence (P-01106–1) encoded by both canonical and alternative transcript (corresponding to position 164 for P-01106–2 isoform, only encoded by the canonical transcript). Interestingly, the mutations were more frequently located in the first 100 aa (36/58 exonic coding SNVs) than in the rest of the protein (P-01106–1) (22/58 SNVs) ($p < 0.0001$, Fisher's exact test). This might be of relevance for common diagnostics, because the immunogen for a widely used antibody (clone Y69) are the first 100aa of the protein. *MYC* gene mutations were also significantly enriched in target motifs of the SHM machinery (RGYW and DGYW motifs, $p < 0.0001$ and $p < 0.0001$, respectively, Fisher's exact test) (Fig. 2a).

Moreover, the coding mutations clustered significantly ($p < 0.0001$, Fisher's exact test) around known phosphorylation site containing regions (using a window of +/− 4 aa positions to evaluate the presence of mutations in proximity to known phosphorylation sites) (Fig. 2b). Since phosphorylation is required for ubiquitination and degradation of the MYC protein by the proteasome, abolition of phosphorylation could lead to a decrease of protein degradation and hence, increase the stability of the MYC protein[31].

Thus, besides IG enhancer hijacking through the IGH-*MYC* translocations, expression of the alternative MYC transcript as

well as protein stabilization due to mutations close to phosphorylation regions might be mechanisms to deregulate activity of MYC in BL.

**IG-non-*MYC* translocations in BL.** Whereas the IG-*MYC* translocations are pathognomonic for BL, IG-translocations involving other commonly affected lymphoma oncogenes, in particular *BCL2* or *BCL6*, are usually absent in BL. Indeed, high-grade B-cell lymphomas with chromosomal breakpoints affecting the *MYC* locus in combination with breakpoints involving the *BCL2* or *BCL6* genes, constitute a separate entity frequently called "DH lymphomas"[1]. Nevertheless, the existence of IG translocations to non-*MYC* partners, which could cooperate with *MYC* in the pathogenesis of BL, has not yet been systematically explored. We discovered in 3/39 cases (8%) IG non-*MYC* translocations by WGS: one case each of IGH-*CBFA2T3* (4170844, solBL), IGH-*HECW2* (4110498, leukBL), and IGK-*CCNG1* (4152036, pleuraBL), considering candidate oncogenes in proximity to the breakpoint on the IG partner chromosomes (Fig. 2a). We validated all three translocations by PCR and Sanger sequencing (Supplementary Data 5). The IGH breakpoint of the IGH-*CBFA2T3* translocation mapped to the IGHA1 switch region, whereas the breakpoint of the corresponding IGH-*MYC* translocation was located in the IGHM switch region. The IGH-*HECW2* translocation had the breakpoint in an IGHG switch region, whereas the IGH-*MYC* translocation affected the IGHM switch region. The IGK-*CCNG1* translocation had breakpoints approximately 350 kb upstream of *CCNG1* and in the IGKV region. Breakpoints in switch regions indicate that CSR was the mechanism leading to the IGH-*CBFA2T3* and IGH-*HECW* translocations, and remarkably again IGHA was involved in one case. It was not possible to determine the timing of the events in the cases with two IGH breakpoints directly from the sequencing data due to the short sequencing reads and the distances of the breakpoint on the IGH locus. However, detailed analysis showed that the IGK-*CCNG1* breakpoint was located exactly at the 3′ end of an IGKV segment and the breakpoint junction contains additional nucleotides (Fig. 3b). Such N-nucleotides are added by the lymphocyte-specific terminal deoxynucleotidyltransferase (TdT) during VDJ recombination mediated by the nucleases RAG1/RAG2. Thus, though rare cases of light-chain revision in germinal center B-cell lymphomas have been described, this event more likely occurred in an early stage of B-cell development in the bone marrow compartment, when the cell attempted to perform an IGKV to IGKJ recombination[32,33]. Overall, this strongly indicates that the IGK-*CCNG1* translocation occurred earlier (at the pre-B or immature B-cell stage) in the evolution of this tumor than the IGH-*MYC* translocation (at the germinal center stage) showing that IG-*MYC* translocations might not always be the initial genetic event in BL (Fig. 3c). All partners of these IG non-*MYC* translocations have been linked to oncogenesis. An IGH-*CBFA2T3* translocation has been previously detected in two patients with pediatric GCB-lymphomas, suggesting that *CBFA2T3* may represent a recurrent oncogene partner of the IGH locus[34]. HECW2 is an ubiquitin-protein ligase that mediates ubiquitination of the tumor suppressor protein *TP73*, a member of the p53 family of transcription factors, and also degrades ATR[35]. Finally, CCNG1 is a cyclin which is thought to play a role in growth regulation, participates in p53-dependent $G_1$-S and $G_2$ checkpoints and might function as an oncogenic protein in ovarian carcinoma[36]. Unfortunately, we could not analyze the expression of the new partners due to lack of RNA in those cases. In summary, oncogenes other than *MYC* can be activated through juxtaposition with IG loci in BL, and some of those events can even precede the IG-*MYC* translocation.

**Fusion transcripts.** Besides oncogene activation by enhancer hijacking as in the case of IG-translocations, the generation of fusion genes is a common mechanism through which SVs can contribute to oncogenesis. Combining the data from WGS and RNA-seq (Supplementary Fig. 7), we identified 58 intragenic fusions at the genome level (excluding the IG and *MYC* loci), eight of which resulted in detectable fusion transcripts (7 intra- and 1 inter-chromosomal) (Supplementary Data 5). The affected genes are involved in key mechanisms of cancerogenesis like proliferation (*TSC22D2* and *ZMAT3*), NF-κB signaling (*BACH2*, *MAP3K7*, and *TRAF3IP2*), JAK-STAT signaling (*CD109*), and immune system regulation (*SOCS5*). We validated the *GTPBP8:C3orf17* and *CD109:TRAF3IP2* fusions at genomic and transcriptomic levels using PCR and Sanger sequencing. Both validated fusions exhibited the genomic breakpoints in the introns of the involved genes (Supplementary Data 5). Interestingly, the GTPBP8:C3orf17 fusion retained the affected introns of both genes in the fusion mRNA. We then investigated the presence of intron retention in all fusion transcripts and identified two additional fusion transcripts, SLC8A1:SOCS5 and TSC22D22:ZMAT3, with evidence of intron retention at the breakpoint junction. Hence 3/8 fusion transcripts (38%) exhibited intron retention at the breakpoint junction, suggesting that gene inactivation by intron retention is a common consequence of fusion gene generation during Burkitt lymphomagenesis.

**Alterations in coding genes.** In recent years, additional genes besides *MYC* have been described as recurrently mutated in BL, usually as a consequence of point mutations[15–17,37]. To elucidate the genes involved in BL lymphomagenesis, we performed an integrative analysis of the different somatic genomic alterations potentially resulting in gene deregulation, namely SNVs, indels, SVs, and copy number aberrations (CNAs). The mean number of gained or lost regions per case was only 2.05 (range 0–10) and 1.82 (range 0–8), respectively; exhibiting a low number of CNAs in accordance with previous cytogenetic and array-based analyses of BL[38] (Fig. 4a). Incorporating all data on genomic alterations, we identified 49 coding genes affected in at least three cases (Fig. 4b, c). Eight of these were affected in ≥20% of samples (*MYC*, *ID3*, *CCND3*, *TP53*, *SMARCA4*, *FBXO11*, *ARID1A*, and *DDX3X*). Selected SNVs and SVs were validated by Sanger sequencing based on DNA availability (Supplementary Data 5 and 6). Correlating the number of mutations per gene to its replication timing in lymphoblastoid B-cell lines showed that many recurrently mutated genes replicate rather early in B-cells (Fig. 4c). Using IntOGen[39], we identified 18 genes as putative driver genes in BL (*ID3*, *TP53*, *CCND3*, *SMARCA4*, *FBXO11*, *ARID1A*, *DDX3X*, *GNA13*, *FOXO1*, *RHOA*, *TCF3*, *PCBP1*, *RFX7*, *E2F2*, *GNAI2*, *TFAP4*, *ADNP*, and *HNRNDP*). All cases exhibited mutations in at least one driver gene. In 29/39 cases (74%) we detected an *ID3* and/or *TCF3* alteration. Disruption of Gα13 signaling via mutations in *GNA13*, *ARHGEF1*, *S1PR2*, *RHOA*, or *P2RY8* has been recently described as a mechanism involved in survival and local confinement of GCB cells to the germinal center[40]. We observed mutual exclusive aberrations of *RHOA* (7/39, 18%), *P2RY8* (7/39, 18%), and *GNA13* (5/39, 13%). Interestingly, *RHOA* or *P2RY8* alterations were observed only in solBL (14/27, 52%) but not in any of the 12 non-solid BL.

**Alterations in non-coding genes.** Recurrent mutations were detected in six non-coding RNAs, namely *RP1–150O5.3* (4/39, 10%), *LINC00939* (3/39, 8%), *RP11–351J23.2* (3/39, 8%), *CASC2* (3/39, 8%), *MIR4447* (3/39, 8%), and *RNU5E-8P* (3/39, 8%) (Fig. 4b). RP1–150O5.3 is located downstream of the *ID3* gene and both genes are co-deleted in the 4 BL cases. The lncRNA

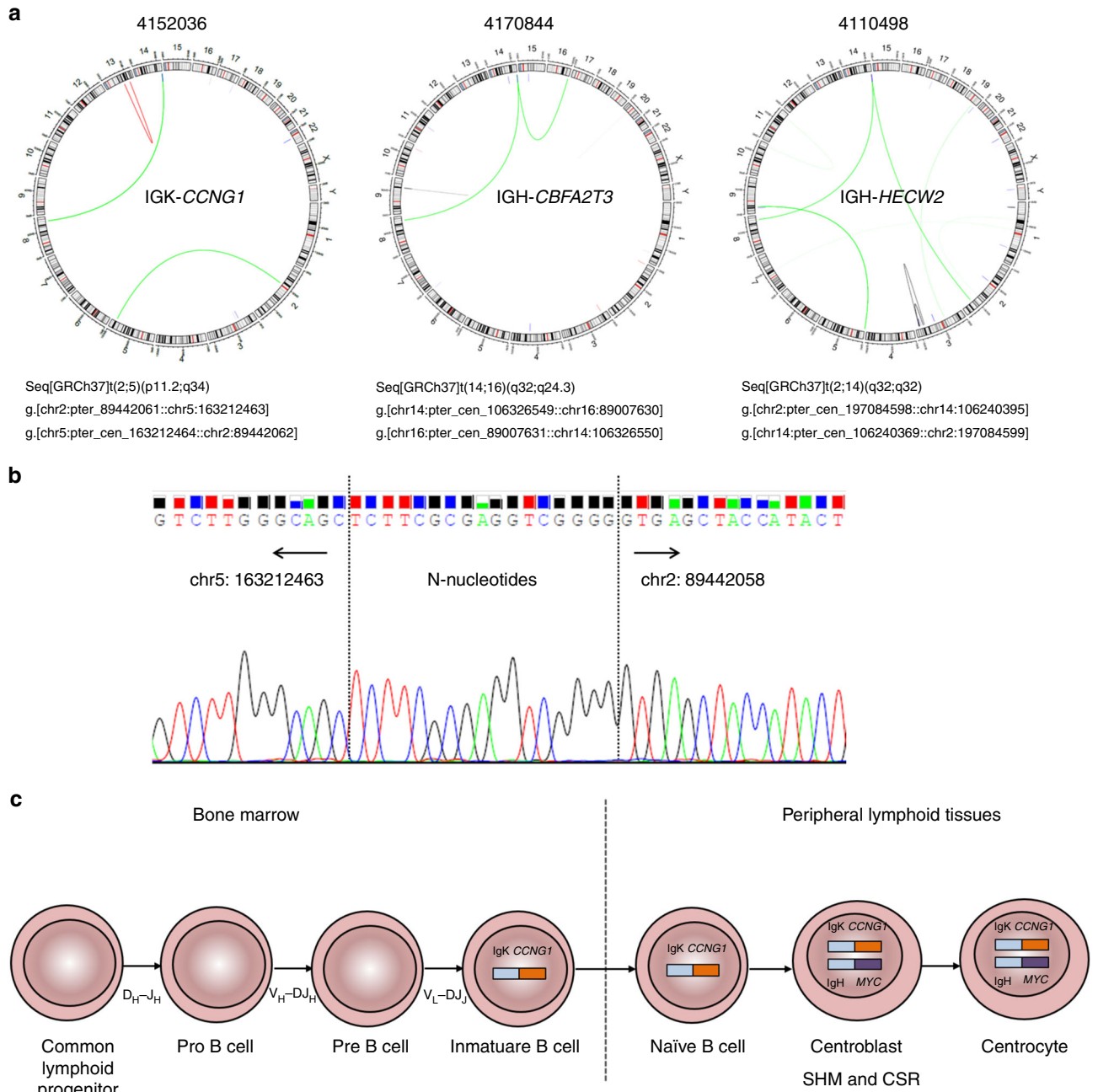

**Fig. 3** IG-non-*MYC* translocations. **a** Circo plots of three BL cases with IG-non-*MYC* translocations; green lines represent translocations, blue lines deletions, red lines duplications, and black lines inversions. The description of the chromosomal translocations affecting IG locus follow the guidelines of ISCN and HGVS nomenclatures using the genomic breakpoints detected by WGS. **b** Electropherogram depicting the junctional sequence of the IGK-*CCNG1* translocation validated using PCR on genomic DNA and Sanger sequencing. The arrows indicate the genomic regions of each chromosomal partner and the discontinued line the exact breakpoint of the translocation. Between both sequences the presence of N-nucleotides was observed. **c** Schematic display of the potential timing of the IGK-*CCNG1* and IGH-*MYC* translocations during BL development in case 4152036. The IGK-*CCNG1* juxtaposition most likely occurred by a mistake during IGK V-J recombination in a B-cell precursor in the bone marrow compartment. The IGH-*MYC* translocation occurred as a later event during tumor evolution due to a mistake of CSR in a GCB cell

cancer susceptibility candidate 2 (*CASC2*), is recurrently deleted in solid tumors, suggesting a tumor suppressor function[41,42]. Recently, *CASC2* was described as being involved in regulation of the PI3K cascade by inhibition of miR-18a, and in regulation of the PTEN pathway by inhibition of miR-21[43]. The *RNU5E-8P* (3q13.31), *MIR4447* (3q13.31), *RP11–351J23.2* (6q27), and *LINC00939* (12q24.32), have to the best of our knowledge not yet been associated with cancer, though RP11–351J23.2 is located in a

candidate tumor suppressor gene locus in lymphomas identified by deletion mapping[44].

**Mutual exclusivity of gene alterations.** To elucidate the role of mutated genes in BL and detect functional modules, we searched for mutual exclusivity of alterations among all recurrently affected coding ($n = 49$) and non-coding genes ($n = 6$) (Fig. 5a). Aberrations of *CCND3* rarely co-occurred with those in the tumor

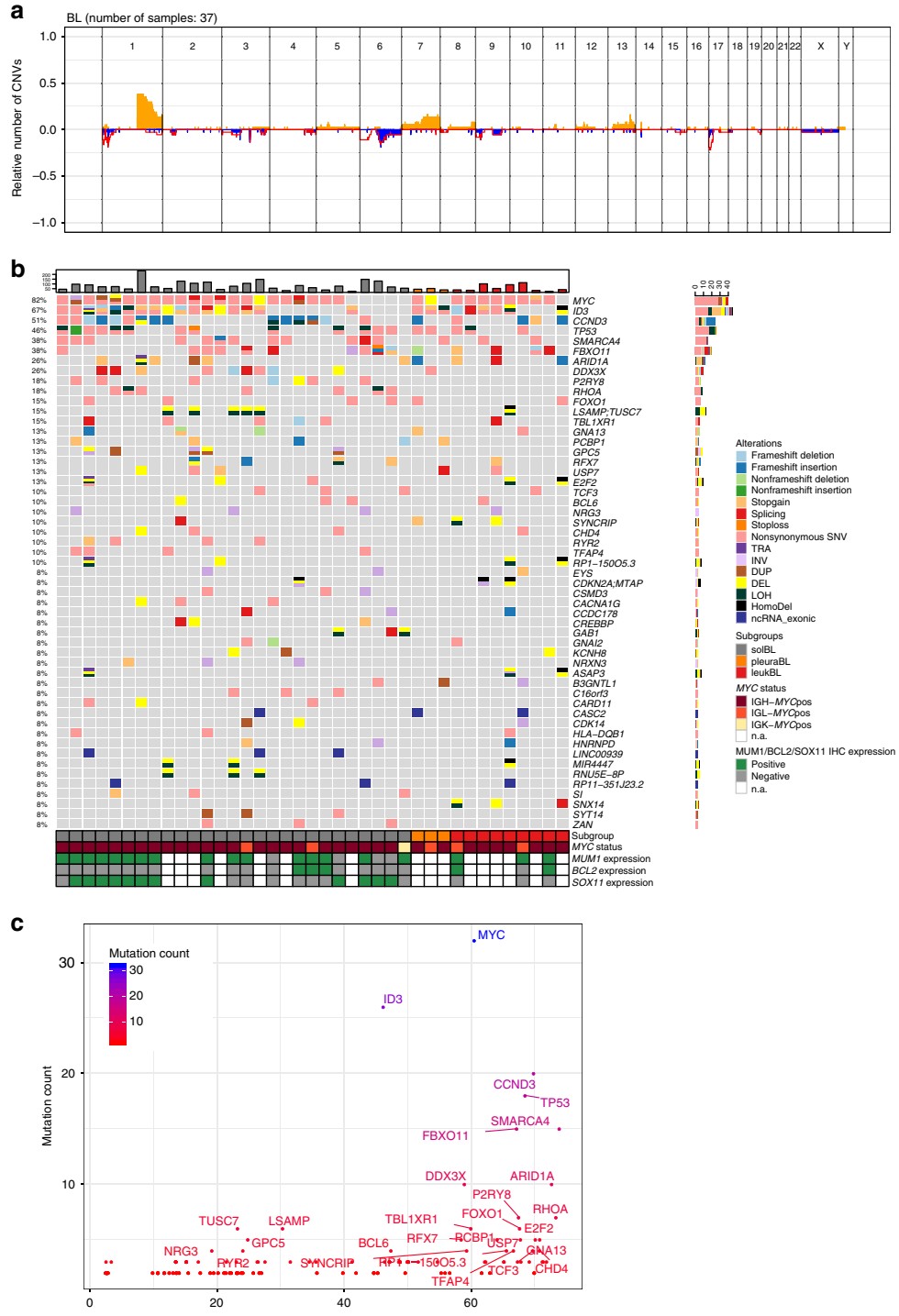

**Fig. 4** Gene dysregulation in BL. **a** Cumulative imbalance profile of 37/39 BL cases excluding the physiological rearrangement in IGH, IGK, and IGL loci. The Fig. does not include the cases with polyploid genome (4108627) and without copy number aberrations (4110996). Losses are indicated in blue, gains in orange, and copy neutral losses of heterozygosity (LOH) as red line. **b** Genes recurrently affected by SNVs, SVs, CNAs, and indels in BL (frequency ≥3 cases). The events in recurrently affected genes are shown independent of the transcript form. The affected genes are ordered by frequency, as well as gene name and the cases are displayed according to the BL subgroups (solBL in gray, leukBL in red, and pleuraBL in orange). The cases are also annotated with MUM1, BCL2, and SOX11 expression status analyzed by immunohistochemistry, as well as *MYC* translocation status. **c** Plot contrasting mutational load and replication time of recurrently altered genes in BL. The *x* axis displays the median replication time in five lymphoblastoid B-cell lines (ENCODE cell lines GM12801, GM06990, GM12812, GM12813, and GM12878) and the *y* axis the mutation count per gene

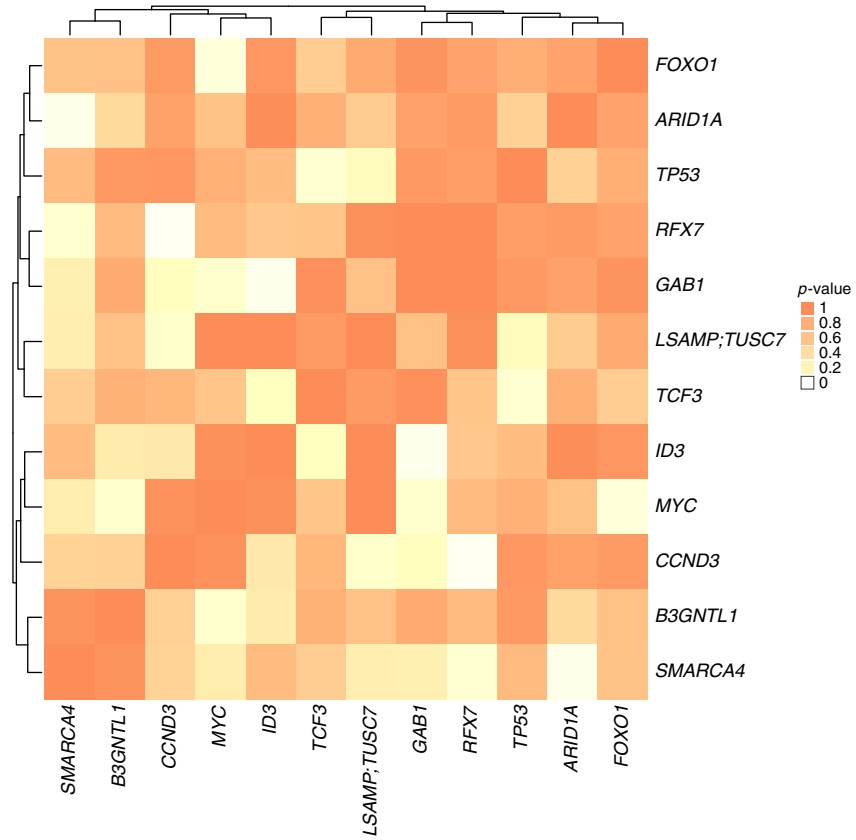

**Fig. 5** Mutual exclusivity analysis. Heat-map showing the mutual exclusivity of recurrently altered genes in BL. The colors indicate the *p*-value of the mutual exclusivity analysis. A low *p*-value indicates that both genes of the respective gene pair are mutated together less frequently than expected

suppressors *LSAMP* and *TUSC7*, and in particular with the transcription factor *RFX7*. Mutual exclusivity argues that there could be a functional connection between RFX7 and CCND3. Although there is no pathway or discrete gene ontology group containing both proteins, there is evidence that both share upstream regulatory genes[45] and that they participate in the same tissue-specific functional interaction network http://hb.flatironinstitute.org/gene/896+64864. Moreover, RFX7 (and other RFX transcription factors) is co-expressed with CCNK, a regulatory subunit of cyclin dependent kinases[46] and RFX7 is described as transcription factor for the oncogene *MYC*. Taken together, these findings suggest that RFX7 could be a transcription factor regulating genes in cell cycle control (including *CCND3*). The precise role of this putative cell-cycle related process in BL will require additional investigation. On the other hand, mutations in *RFX7* were also to a considerable extent exclusive to *SMARCA4*, which in turn showed strong exclusivity with mutations in yet another member of the SWI/SNF complex, *ARID1A*. Alterations in the *ID3* gene were mutually exclusive with alterations of *TCF3* and also *GAB1*, which are all implicated in tonic BCR signaling[47]. Remarkably, *FOXO1* and *MYC* mutations were also to a considerable extent exclusive. A mutual exclusivity between *TP53* and *CDKN2A* alterations has been previously described in large B-cell lymphomas and BL[48], however, using our approach and considering *p*-value <0.1 as cut-off for mutual exclusivity, we did not identify this association in our series. Manual inspection revealed that indeed, alterations of *TP53* and *CDKN2A* seem to occur mutually exclusive. Moreover, the cases exhibiting genomic aberrations in *TP53* showed higher expression of CDNK2A (*p*-value <0.1) (Supplementary Fig. 7b). Overall, the integrative analysis revealed that mainly three

interacting biological pathways or complexes are altered in BL: proliferation and survival, the SWI/SNF complex, and tonic BCR signaling.

**Differential splicing of genes altered in BL**. To investigate if BL exhibit differential splicing as complementary mechanism to alterations affecting the coding sequence on genome level, we analyzed differential splicing in 77 genes including all predicted driver genes, all recurrently (≥3 cases) mutated genes, all new IG non-*MYC* translocation partners, and all genes involved in fusion transcripts identified in the present study. We compared the RNA-seq based expression pattern to normal GCB cells and identified differential splicing in 40/77 genes (52%). Among those was *TCF3*, which showed upregulation of the isoform E47 and downregulation of E12 in BL as compared to GCB cells (Fig. 6a). At the genomic level the mutations in *TCF3* observed in our cohort and described previously[17,37] are located in the helix-loop-helix (B-HLH) DNA binding and dimerization domain of the E47 splice isoform, but remarkably not in the E12 encoding part of the gene. We correlated the expression of both TCF3 isoforms to the mutational status of its negative regulator *ID3*. The relative expression of the isoform E47 showed a strong trend towards higher expression (*p* = 0.08, Wilcoxon rank sum test) in the BL cases without *ID3* mutations (Fig. 6b). The protein isoforms E47 and E12 are predicted to differ in the electrostatic charge of the B-HLH domain which might affect DNA binding. The isoform E47 contains 4 negatively charged residues less than isoform E12, showing an overall more positive charge at the DNA-binding groove (Fig. 7a–c). Thus, increased expression of E47 might similarly enforce TCF3 binding activity to DNA like inactivating

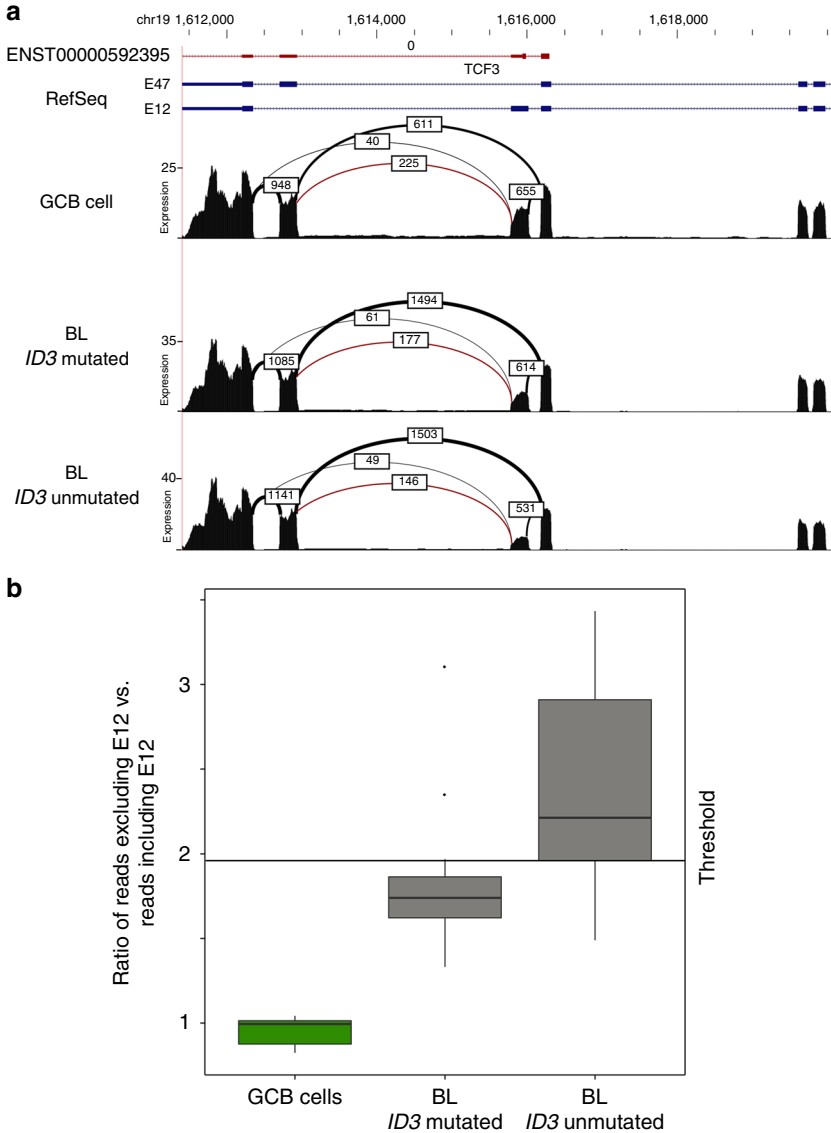

**Fig. 6** TCF3 splicing. **a** Sashimi plot showing the expression of the 3′ end of TCF3. On the top, the ENST00000592395.5 ensemble isoform displaying both alternative exons is shown. Below, the RefSeq annotations for the isoforms E47 and E12 are shown. Each isoform includes only one of the alternative exons. The three Sashimi plots show the expression and spliced reads of GCB cells (top), BL with *ID3* mutations (center), and BL without *ID3* mutations (bottom). The numbers indicate the amount of spliced reads between the connected exons. The red arc marks the isoform that includes both alternative exons. Note the slight difference in expression levels between the three plots. The cassette exon of the E12 isoform shows a decrease in inclusion (and expression) from GCB cells (48%) to BL with *ID3* mutation (29%) to BL without *ID3* mutations (26%). **b** Ratio of spliced reads that directly go to the E47 exon vs spliced reads that go to the E12 exon starting from the right one of the two central exons shown in Fig. 5a. While half of the spliced reads go to E47 in control (green), most of the *ID3* unmutated BL cases have more than two thirds of the spliced reads going to E47 (right box) ($p = 0.070$) (t test, two sided). In *ID3* mutated BL cases ($n = 11$) (central box), this number lies between the control and the *ID3* unmutated cases ($n = 9$) ($p = 0.09$) (Wilcoxon rank sum test)

its negative regulator *ID3* by mutations or introducing activating mutations in *TCF3*.

We extended the analysis of the differential expression of the E47 and E12 TCF3 isoforms to adult GCB-lymphomas other than BL. To this end, we mined RNA-seq data from FL ($n = 85$), DLBCL ($n = 72$), and FL-DLBCL ($n = 17$) generated within the ICGC MMML-Seq consortium using the same sequencing pipeline (unpublished data). In this non-BL cohort the preferential usage of the isoform E47 is significantly lower than in BL (BL vs non-BL, $p < 0.001$) (Supplementary Fig. 8). Given that a recent study has identified differential expression of the E12 and E47 TCF3 isoforms in pluripotent human embryonic stem cells (ESCs) compared to differentiated cells we extended our analysis to published data from ESCs and induced pluripotent stem cells

(iPSCs)[49,50] (Supplementary Fig. 8). Indeed, we corroborated the relatively high expression of E12 as compared to a relatively low expression of E47 in ESCs and iPSCs cells (joined ESCs/iPSCs vs GCB cells $p = 0.002$, Wilcoxon rank sum test). Thus, differential expression of the E47 and E12 TCF3 isoforms seems to be a common mean to regulate TCF3 function. Skewing towards the E47 expression seems to be particularly pronounced in BL as compared to other mature GCB-lymphomas and, thus, likely contributes to deregulation of the TCF3/ID3 complex particularly in BL lacking *ID3* and/or *TCF3* mutations.

Another gene showing significant differential splicing in BL as compared to GCB cells was *CBFA2T3*. Here, expression of isoform 2 which lacks amino acids 1–61 and 102–126 of the canonical form was preferentially expressed in BL. Isoform 2 acts

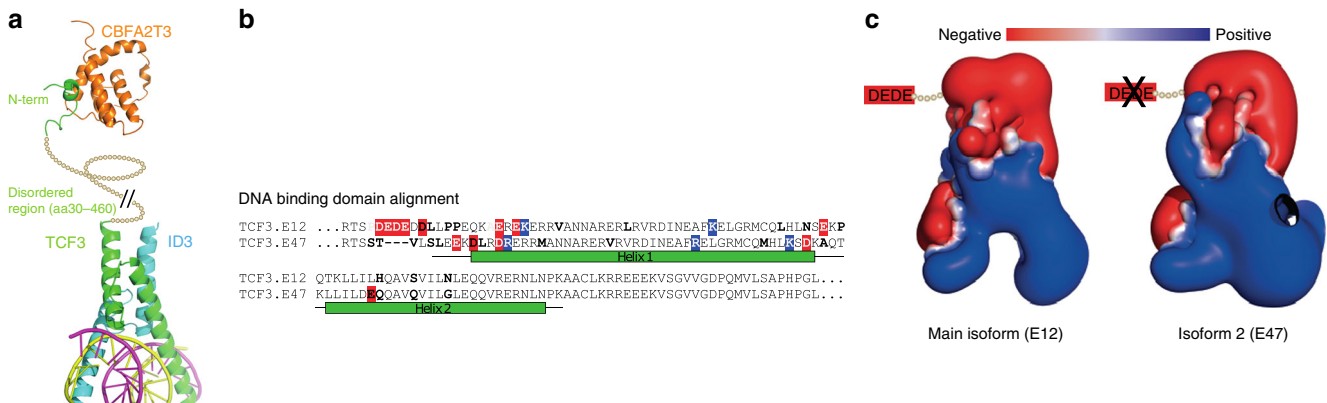

**Fig. 7** The ID3-TCF3-CFBA2T3 complex. **a** Protein structure of TCF3 (green) in complex with ID3 (blue) and bound to DNA (bottom). The N-terminal region that interacts with CBFA2T3 (orange) is separated by a long disordered/unstructured region (shown as a looped line). **b** Alignment of the DNA-binding region (residues 527–644) of TCF3 (Uniprot: P15923). The location of the domain is inferred by sequence similarity to the structure from protein databank code 2ypa chain A (a heterodimeric complex involving an equivalent domain from human TAL1 in complex with DNA). Differences between the sequences are highlighted (uncolored and bold for uncharged amino acids; red for negatively charged and blue for positively charged amino acids). Note that many changes alter the charge in one isoform relative to the other, leading to a net gain of four negative charges in isoform E12 relative to isoform E47. **c** Representations of the electrostatic surface of the E12 (left) and E47 (right) isoforms. The surface of E47 has a more positive electrostatic potential (blue). Also indicated is the loss of four negative (Glu/Asp) residues at the N-terminal portion of the domain, which were not present in the model (as there was no suitable structural template for them). The loss of these four negative charges also means that the E47 isoform is relatively more positively charged

as an A-kinase-anchoring protein, interacting with PDE7A and PRKAR2A[51], and thus regulates cAMP-mediated signaling. This pathway is regulated by GNAS-coupled G-protein coupled receptors, for which mutations in colorectal cancer have been shown to activate cAMP signaling[52]. In order to investigate the mechanism causing this differential splicing, we mined our previously published DNA methylation data[19], and identified a region in *CBFA2T3* differentially methylated (DMR) between BL and GCB cells. DNA methylation of this DMR is negatively correlated with gene expression of the long transcript (Fig. 8). Interestingly, this DMR in *CBFA2T3* contains a transcription factor binding site for TCF3 (based on ENCODE data), suggesting that TCF3 might be involved in the regulation of alternative splicing of CBFA2T3[19]. Moreover, the IG translocation associated breakpoints in *CBFA2T3* in the case described above and in another, previously published pediatric mature B-cell leukemia diagnosed morphologically as B-AL[34] disrupts isoform 1 and should lead to preferential expression of isoform 2. In acute myeloid leukemia, *CBFA2T3* can be a translocation partner of *RUNX1*, producing a chimeric protein with breakpoints usually between exons 1 and 2, or exons 3 and 4 of *CBFA2T3*[53]. TCF3 binds both CBFA2T3[54] and ID3[55] likely via N- and C-terminal regions, respectively[56] (Fig. 7a). This suggests that CBFA2T3, which is dysregulated in BL, via translocation, differential splicing, and differential methylation, belongs to an *ID3-TCF3-CBFA2T3* network important to Burkitt lymphomagenesis.

**Germline mutations.** Several germline mutations predisposing individuals to particular cancers are known, but to date only few have been described for BL[57]. Aiming at clearly damaging changes we focused on protein-truncating germline mutations in 114 known cancer predisposition genes in the present BL cohort[58]. We found such germline mutations in *CHEK2* and *FANCG* each in one, and two germline mutations in the *BLM* gene in another patient. All germline mutations were validated by PCR and Sanger sequencing (Supplementary Data 6). Although a predisposing role of the heterozygous *FANCG* for BL could not firmly be established, we consider the changes in *CHEK2* and

*BLM* relevant for the pathogenesis of BL. The frameshift mutation in *CHEK2* leads to a truncated protein (p.Thr410MetfsTer15). Germline mutations in *CHEK2* (OMIM: 604373), in particular truncating mutations, are considered causative of Li Fraumeni (like) syndrome 2 (LFS2) (OMIM: 609265) and are associated with increased cancer risk, with lymphoma belonging to the disease spectrum of LFS[59,60]. The *BLM* mutations were compound heterozygous, leading to biallelic inactivation. Indeed, clinical features of the patient were in line with the phenotype of Bloom syndrome (OMIM: 210900) but the diagnosis was not made prior to the current study. Individuals with Bloom Syndrome, which is a rare autosomal recessive disorder characterized by genomic instability, have been described to carry a higher risk of developing cancers including lymphomas. Taken together, 2/39 (5%) patients with pediatric BL investigated in this series carry likely pathogenic germline changes predisposing to this disease.

In addition to the known cancer predisposition genes we also investigated the presence of germline mutations in genes showing somatic alterations in BL. Focusing on the same 77 genes that were subjected to the differential splicing analysis, we identified germline nonsynonymous SNVs, non-frameshift insertions or splice site mutations in 26 of 77 (34%) genes analyzed (27/39 patients), with frequencies from 3 to 13% (Fig. 9a). We observed co-occurrence of germline and somatic mutations in the same gene in two patients, indicating double hits affecting these genes (*FOXO1* and *RYR2*). After filtering for allele frequency of the mutations in a public database (gnomAD, 11.09.2017), the impact of the variants at protein level, the tolerance/intolerance of a gene towards new functional mutations, and the presence of the variant in the non-BL cohort from the ICGC MMML-Seq project, we ended up with 6 germline mutations across 5 genes affecting 6 patients that could be potentially relevant for lymphomagenesis. The genes affected were *NCOR1*, *CREBBP*, *RYR2*, *ARID1A*, and *NRXN3*.

In total, 72% (28/39) of BL cases exhibited germline mutations in cancer predisposition genes or/and genes somatically mutated in BL, indicating a potential crucial contribution of germline events in BL lymphomagenesis.

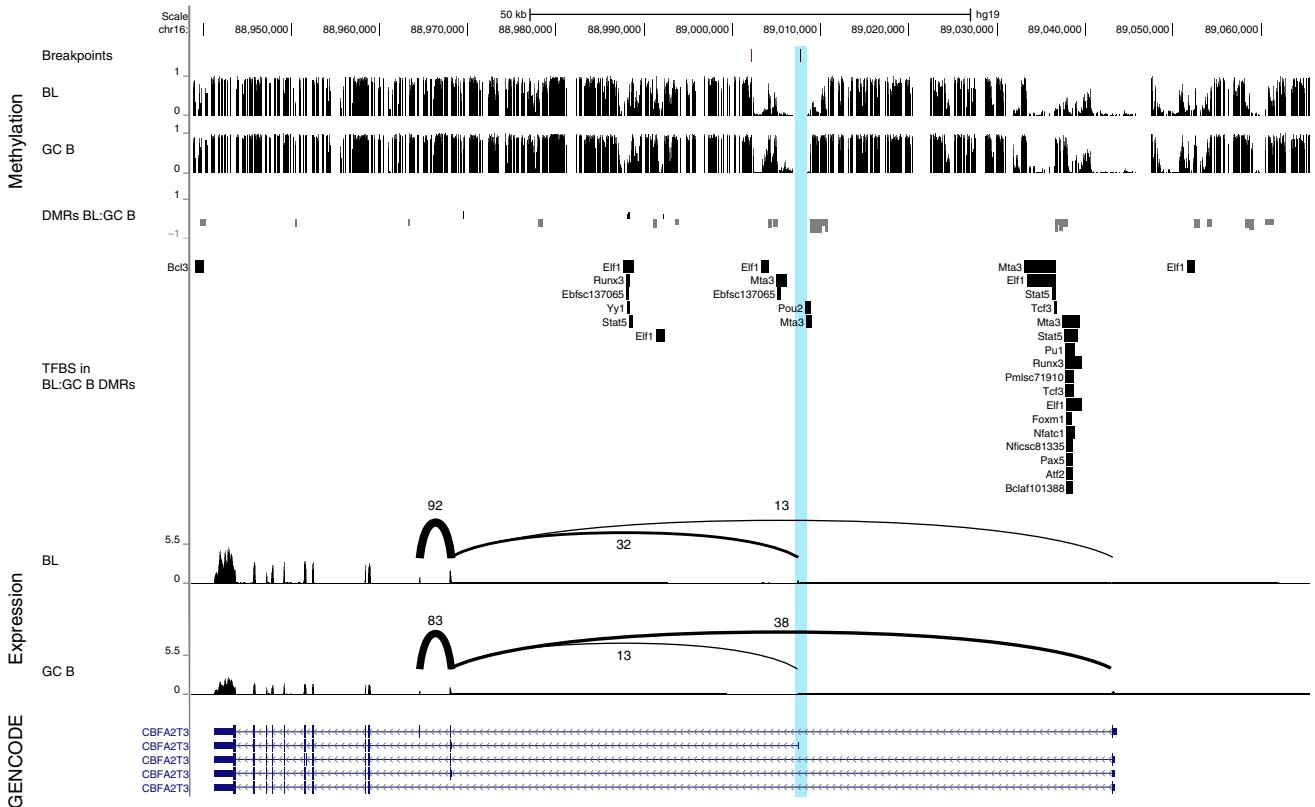

**Fig. 8** Integrative analysis of breakpoint location, DNA methylation, and expression of *CBFA2T3*. Composite plot showing breakpoint locations (top), DNA methylation (second from top), transcription factor binding sites (TFBS, middle) and observed splicing patterns (second from bottom) of BL and GCB cells at the *CBFA2T3* locus. *CBFA2T3* is significantly differentially spliced between BL and GCB, with BL predominantly expressing the shorter isoform 2. The TSS of isoform 1 preferred by GCB overlaps with a herein described breakpoint (black mark in blue highlight) and is proximal to a specific breakpoint (red mark) which was previously identified in pediatric mature B-cell leukemia (diagnosed morphologically as B-AL)[34]. In addition, several DMRs between BL and GCB are located in the *CBFA2T3* locus[19]. A promoter associated negative cDMR (negative correlation between DMR methylation and gene expression) overlaps, inter alia, with a TCF3 transcription factor binding site, suggesting a regulatory role of TCF3 in CBFA2T3 isoform expression

**Mutational signatures.** Germline mutations in some cancer predisposing genes have been linked to certain mutational signatures in the tumor cells. To investigate whether the patients with the *BLM* and *CHEK2* mutations showed particular mutational patterns, we explored the distribution of SNVs over the genome in all BL. We observed four regions of increased SNV density in BL with at least five SNVs with at most 1000 bp intermutational distance, similar to what has previously been defined as Kataegis[61]. These regions were restricted to the *MYC* and the three IG loci (Fig. 9b). Remarkably and in contrast to other GCB-lymphomas we did not identify further regions of recurrent kataegis. Applying a new mathematical approach for supervised mutational signature analysis (unpublished data), we identified 6 of the 30 previously described signatures (AC1—spontaneous deamination, age related, 32/39 = 82.1% of the samples affected; AC2—APOBEC, 27/39 = 69.2%; AC6—mismatch repair defects, 39/39 = 100%; AC9—mistakes in polymerase η 39/39 = 100%; AC10—mistakes in polymerase ε, 38/39 = 97.4%; and AC17—mechanism unknown, 28/39 = 71.8%)[62,63] (Fig. 9c). Furthermore, we identified contribution by three new mutational signatures (L1–39/39 = 100%, L2–38/39 = 97.4%, and L3–16/39 = 41.03% of the samples affected, respectively), discovered in a larger cohort of GCB-lymphomas in the framework of ICGC MMML-Seq (unpublished data). Two of these latter signatures may be attributed to the action of AID; one with a low degree of modulation by altered repair pathways and associated with CSR (L1) and one with a high degree of modulation and associated with SHM (L2). Remarkably, the median contribution of the signatures L1 and L2 to the overall mutations was 30% and 21%, respectively, indicating that around 50% of all mutations in BL are related to the B-cell specific mechanisms CSR or SHM. Thus, we establish a link between mutational processes active in GCB cells and BL lymphomagenesis. However, we did not find the patients with germline mutations to differ in the mutational patterns of the tumor cells.

## Discussion

In the present work, we used up-to-date integrated sequencing analyze to study the mechanisms underlying Burkitt lymphomagenesis. In depth characterization of IG-*MYC* breakpoint sequences, the *MYC* mutations, and MYC transcripts allowed us to reassess the mechanisms leading to generation of the IG-*MYC* translocations and to identify a complex interplay of mutational, regulatory, transcriptional, and possibly post-transcriptional mechanisms leading to enhanced MYC activity. This very much underscores the central role for MYC in BL pathogenesis and how pathogenetic alterations at various levels contribute to the deregulation of MYC activity. We also revealed a series of mechanisms cooperating with MYC in lymphomagenesis, caused by a wide variety of changes, including germline mutations, somatic mutations, and structural aberrations associated with enhancer hijacking and intron retention as well as alternative splicing.

We have shown, that IG-*MYC* translocations in sBL are generated mostly via aberrant CSR or less frequently by SHM, and

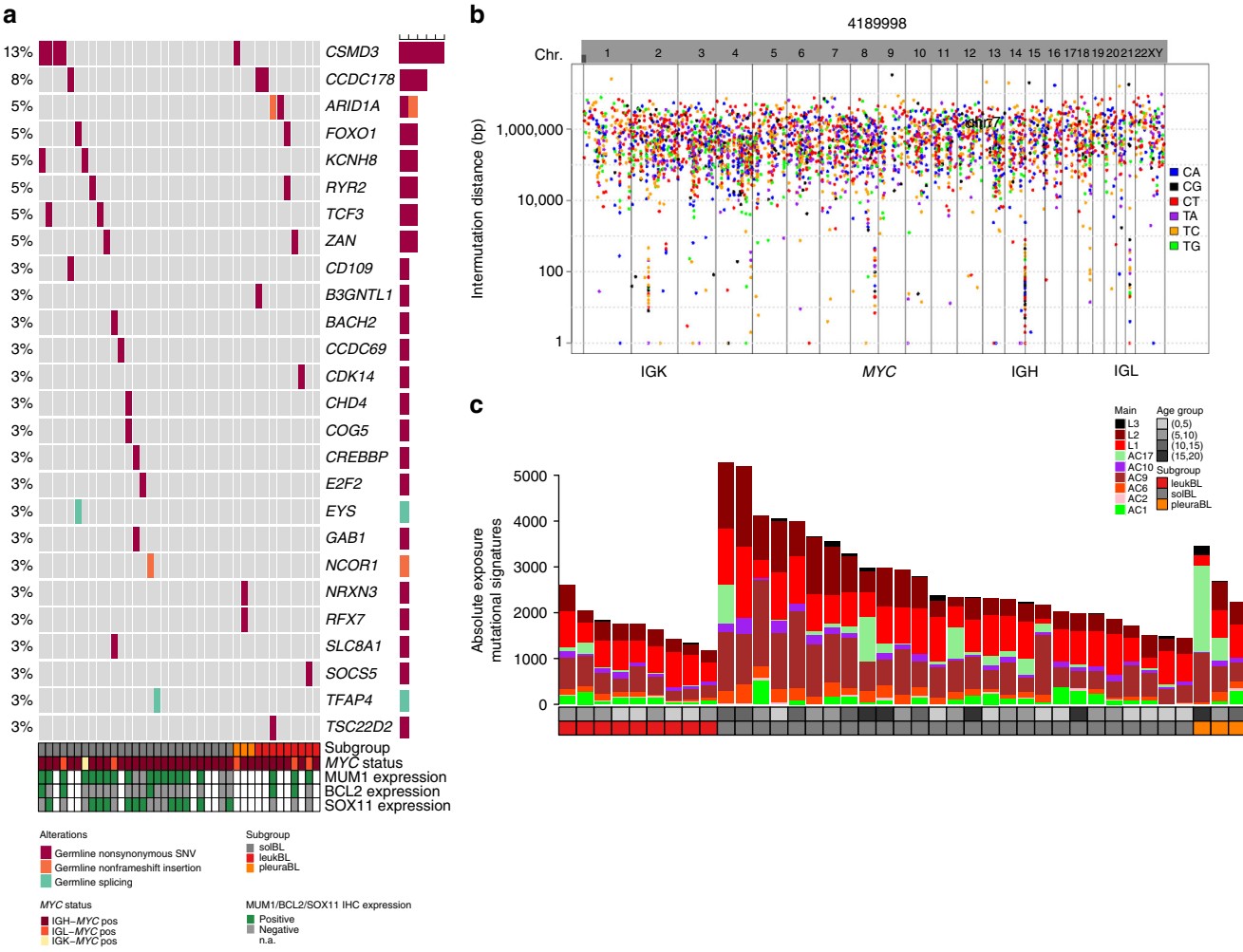

**Fig. 9** Germline events and mutational signatures. **a** Germline mutations in genes recurrently altered by somatic mutations in BL. The affected genes are ordered by the incidences as well as names and the BL cases are displayed according to their subgroups (solBL in gray, leukBL in red, and pleuraBL in orange). Moreover, the BL cases are annotated with the MUM1, BCL2, and SOX11 expression status determined by immunohistochemistry, as well as MYC translocation status. **b** Rainfall plot of a representative BL case (4189998). The nucleotide changes are labeled using different colors. The x axis encodes the genomic coordinate and the y axis the log-scaled intermutation distance. Clusters of hypermutation (Kataegis) can be identified as "rainfalls" reaching very low intermutation distance. As in this representative case, in the other BL cases, the IG and MYC loci exhibited a high density of SNVs. **c** Absolute exposures of mutational signatures in the BL samples extracted from combined supervised and unsupervised analyses of mutational signatures. The mutational signatures are labeled with different colors and the BL cases are displayed according to their subgroups (solBL in gray, leukBL in red, and pleuraBL in orange). The cases are also annotated for the age group. Heights of the stacked bar plots correspond to the number of SNVs associated to the respective mutational signatures

that IGHA breakpoints were almost exclusive for BL in our large series of GCB-lymphomas. These findings extend previous targeted studies using Southern blot or PCR-techniques[64,65] showing recurrent MYC translocation breakpoints in IGHA in BL and suggesting that the cell of origin of sBL is a germinal center experienced B-cell primed to switch to and, thus, expressing IgA. This conclusion is also supported by the clinical observation that sBL frequently presents in lymphatic tissue of the ileocoecal region, i.e., one of the sites of the human body particularly important for IgA production[1,66].

The overlap of affected genes in pediatric sBL on the one hand and in other GCB-lymphomas occurring in children like pediatric DLBCL[67] or pediatric-type FL[68] on the other hand is small (Supplementary Fig. 9). In contrast, there is a considerable overlap with regard to genes affected in eBL despite some differences in mutation frequencies[18,69] (Supplementary Fig. 10). Thus, although sharing a development in young children and a derivation from germinal center B-cells, pediatric sBL, DLBCL

and FL show highly distinct pathogenesis pathways. On the other hand, pediatric sBL and eBL, although developing in different continents and having distinct key co-factors for their pathogenesis (association with malaria infection of the children and much higher frequency of tumor cell infection with EBV in eBL) show a similar landscape of genetic lesions, and thus a closely related pathogenesis.

Finally, despite the considerable variety in dysregulating mechanisms the targeted pathways and complexes seem to be rather conserved and converged on few cellular functions including proliferation and survival, the SWI/SNF complex, and tonic BCR signaling. This seemingly functional simplicity on the basis of a marked aberrational complexity renders BL a good candidate for both ex vivo modeling and in vivo targeted therapy.

## Methods

**Experimental model and subject details**. Methods and procedures applied by the MMML and ICGC MMML-Seq have been detailed in various publications[16,19,70]

of the networks and are summarized in the following section. Moreover, the experimental as well as computational procedures for the analysis of WGS and transcriptome data are also described in the uncommitted manuscript by Hübschmann et al., under revision.

The ICGC MMML-Seq cohort comprises pre-treatment tumor tissue and corresponding matched normal material (peripheral blood, buffy coats without clonal IGHV rearrangement) obtained with informed consent of the respective patients and/or in minors their legal guardian. In addition, sorted germinal center B-cells (GCB) and naive B-cells from non-neoplastic tonsils were included. Both normal cell populations were obtained by flow cytometry immunophenotyping sorting (FACS) using specific markers for GCB (CD20, CD23, CD27, and CD38) and naive B-cells (CD20hi and CD38). The ICGC MMML-Seq study has been approved by the Institutional Review Board of the Medical Faculty of the University of Kiel (A150/10) and of the recruiting centers.

Tumor samples were reviewed by expert hematopathologists and classified according to the WHO 2008 guidelines. A consensus diagnosis was achieved by a single independent microscope analysis if at least five of seven pathologists agreed by discussion. For discrepant cases, a consensus was obtained after discussing the respective cases at a multiheaded microscope. The immunophenotypic and morphologic characteristics were evaluated on formalin-fixed and paraffin-embedded tissue sections of the diagnostic tumor biopsies. The immunophenotype data was obtained using an immunohistochemical panel including antibodies directed against CD20, CD10, BCL2, BCL6, and MUM1/IRF4. Lymphoma samples were scored according to their percentage content of positive tumor cells from 0 to 4(0 = 0%; 1 = 1–25%; 2 = 26–50%; 3 = 51–75%; 4 = 76–100%). Immunohistochemical staining for Ki-67 was assessed as the percentage of positive tumor cells on FFPE material. In addition, in situ hybridization staining for Epstein–Barr encoding region was done in tissue sections. Moreover, in the cases with leukemic presentation, the immunophenotype data was obtained using flow cytometry exploring the same immunophenotype markers as described above.

FISH on interphase nuclei was performed on frozen tissue sections applying the specific probes LSI BCL6, LSI MYC, LSI IGH/MYC, CEP8 Tricolor, LSI IGH, and LSI BCL2. All of the probes were provided by Abbott Molecular Diagnostics. Digital image acquisition, processing, and evaluation were performed using ISIS digital image analysis version 5.0 (MetaSystems, Altussheim, Germany). The signal distribution was evaluated by two independent observers.

In the framework of the ICGC MMML-Seq network, we included in the present study samples of sporadic BL in children using the following inclusion criteria: diagnosis of BL according to the WHO 2008 criteria[71], age at diagnosis ≤18 years, presence of an IG-MYC rearrangement detected by FISH and/or WGS and absence of chromosomal translocations affecting the BCL2 and/or BCL6 genes by FISH and WGS, i.e., so called single hits[72] (Supplementary Fig. 1a and Supplementary Data 1). Using these criteria a total of 39 patients entered this study. Seventeen cases were previously published[16,19,70,73]. All patients were registered in one of the clinical trials of the Mature B-cell-non-Hodgkin-Lymphoma, Berlin-Frankfurt-Münster (NHL-BFM) study group and treated according to respective protocols: 38 patients in the B-NHL BFM 04 and one in the B-NHL-BFM 95 trial[74]. Based on available clinical, virological, and serological data all cases were described to be HIV- and EBV-negative. Analysis of WGS and RNA-seq data confirmed this, except for one tumor sample showing EBV sequences by WGS (but not by RNA-seq or EBER in situ hybridization) and one matched normal (blood) sample (but not the corresponding tumor samples) showing traces of HIV1 virus sequences (Supplementary Data 1).

**Sample processing**. The study was performed in accordance with the ICGC guidelines (www.icgc.org). The experimental procedures for DNA and RNA extraction, the detection and sequencing of immunoglobulin have been published previously[16,19]. The tumor cell content in the cryopreserved sample material was estimated to be at least 60% in all cases.

**Sequencing**. DNA libraries of the tumor and matched normal samples were prepared using the TruSeq DNA library Preparation Kit Sets A and B (Illumina; estimated insert size of 343 bp) or TruSeq Nano DNA library Preparation Kit (Illumina; estimated insert size of 350 bp). Clusters were generated with cBot and the TruSeq PE Cluster Kit v3 cBot HS (15023336_A, Illumina). Paired-end sequencing was performed on Illumina HiSeq2000 (2x 100 bp), HiSeq2500 (2x125 bp), or Hiseq-X10 (2x 150 bp) instruments using the TruSeq SBS Kit, Version 3 (200 cycles).

RNA libraries of the tumor samples and sorted germinal center B-cells from non-neoplastic tonsils were prepared using the TruSeq RNA library preparation Kit Set A and B, at an insert size of ~300 bp according to manufacturer´s instructions. Two barcoded libraries were pooled per lane and sequenced on Illumina HiSeq2000 or HiSeq2500.

**Whole-genome sequencing data processing**. Read pairs were mapped to the human reference genome (build 37, version hs37d5, (ftp://ftp.1000genomes.ebi.ac.uk/vol1/ftp/technical/reference/phase2_reference_assembly_sequence/hs37d5.fa.gz), using bwa-mem (version 0.7.8 with minimum base quality threshold set to zero [-T 0] and remaining settings left at default values)[75], followed by coordinate-

sorting with bamsort (with compression option set to fast (1)) and marking duplicate read pairs with bammarkduplicates (with compression option set to best (9)) (both part of biobambam package version 0.0.148).

Somatic SNVs and indels in matched tumor normal pairs were identified using the DKFZ core variant calling workflows of the ICGC PCAWG project (https://dockstore.org/containers/quay.io/pancancer/pcawg-dkfz-workflow). Initial candidate variants for SNVs in the tumor were generated by samtools and bcftools (version 0.1.19), followed by a lookup of the corresponding positions in the control. To enable calling of variants with low allele frequency we disabled the Bayesian model (by setting -p 2). Thus, all positions containing at least one high quality non-reference base are reported as candidate variant. The resulting raw calls were categorized into putative somatic variants and others (artifacts, germline) based on the presence of variant reads in the matched normal sample. The frequency of all putative somatic variants was then refined by checking for potential redundant information due to overlapping reads and precise base counts for each strand were determined. All variants were annotated with dbSNP141, 1000 Genomes (phase 1), Gencode Mapability track, UCSC High Seq Depth track, UCSC Simple-Tandem repeats, UCSC Repeat-Masker, DUKE-Excluded, DAC-Blacklist, UCSC Selfchain. The confidence for each variant was then determined by a heuristic punishment scheme taking the aforementioned tracks into account. In addition variants with strong read biases according to the strand bias filter were removed. High-confidence variants were used for further analysis. To identify indel events tumor and matched control samples were analyzed by Platypus[76] (version 0.8.1). All variants indicating an indel were categorized into putative somatic and other based on the genotype likelihoods (matched genotype 0/0 for somatic indels). High-confidence somatic variants were required to either have the Platypus filter flag PASS or pass custom filters allowing for low variant frequency using a scoring scheme. Candidates with the badReads flag, alleleBias, or strandBias were discarded if the variant allele frequency was <10%. Additionally, combinations of Platypus non-PASS filter flags, bad quality values, low genotype quality, very-low variant counts in the tumor, and presence of variant reads in the control were not tolerated. In order to remove recurrent artifacts and misclassified germline events, somatic indels that were identified as germline in at least two patients in the ICGC MMML-seq cohort were excluded. For two samples (4152036, 4178518) tumors and their matched controls were sequenced on different Illumina instruments (Hiseq2500 and HiseqX). To prevent technology-specific artefacts, the standard SNV and indel calling workflow was extended with filters developed for samples without matched control. To this end, variants recorded in dbSNP version 147[77] with "COMMON = 1" tag were removed, but rescued if they had a corresponding OMIM record in dbSNP. We additionally removed mutations found in ExAC version 0.3.1 (>0.1%; Lek et al., 2016), EVS (>1%; Exome Variant Server, NHLBI Exome Sequencing Project (ESP)) and our control dataset (>2%, among 280 controls). Somatic small variants in these samples were further filtered out if their respective position was covered insufficiently in the control sample (<20X) or if the fraction of variant reads in the control was too high (>1/30).

For some samples (4100314, 4103570, 4163741, 4170844, 4190231, 4190316) increased SNV artefact rates were detected, which were related to higher base quality scores for wrongly called bases. For these samples the base quality threshold was increased from 13 to 20 and low mutant allele frequency (MAF) penalty was switched off (i.MPILEUP_OPTS="-REI -Q 20 -q 30 -ug"; CONFIDENCE_OPTS="  -c 0 -l 1").

SNVs and indels were annotated using ANNOVAR[78] according to GENCODE gene annotation (version 19) and overlapped with variants from dbSNP (build 141) and the 1000 Genomes Project database. SNVs classified as splicing, nonand accurate short read alignment with Burrowssynonymous changes, stop-gains, and stop-losses were predicted to affect protein function.

**Detection of copy number alterations and allelic imbalances**. Allele-specific copy number alterations were detected using ACEseq (allele-specific copy number estimation from whole-genome sequencing; unpublished data).

ACEseq determines absolute allele-specific copy numbers as well as tumor ploidy and tumor cell content based on coverage ratios of tumor and control as well as the B-allele frequency (BAF) of heterozygous single-nucleotide polymorphisms (SNPs). SVs called by SOPHIA were incorporated to improve genome segmentation.

Ploidies were manually checked and compared with FISH results. Adjustments were made if necessary. Accordingly, tumor cell content estimates were compared to the doubled median MAF and adjusted in case ACEseq and MAF-based estimates deviated by more than 10% from each other.

Final copy number segments were further smoothed to calculate the total number of gains and losses. Neighboring segments were merged if they rounded to the same copy number and deviated by less than 0.5 copies in case of segments <20 kb or deviated by less than 0.3 copies otherwise. Remaining segments <500 kb were merged with their closer neighbor based on allele-specific and total copy number and once again segments smaller than 2 Mb deviating by less than 0.4 copies were merged. Based on the resulting segments the number of gains and losses was estimated.

Furthermore, the fraction of aberrant genome was estimated corresponding to the fraction of the genome that is classified either as duplication or deletion (>0.7 deviation from the ploidy) or was identified as a loss of heterozygosity.

**Detection of genomic structural rearrangements.** SV were called using the SOPHIA algorithm (unpublished data). The source code of SOPHIA is available at https://bitbucket.org/utoprak/sophia/. Briefly, SOPHIA uses supplementary alignments as produced by bwa-mem as indicators of a possible underlying SV. SV candidates are filtered by comparing them to a background control set of sequencing data obtained using normal blood samples from a background population database of 3261 patients from published TCGA and ICGC studies and both published and unpublished DKFZ studies, sequenced using Illumina HiSeq 2000, 2500 (100 bp), and HiSeq X (151 bp) platforms and aligned uniformly using the same workflow as in this study. An SV candidate is discarded if (i) it has more than 85% of read support from low quality reads; (ii) the second breakpoint of the SV was unmappable in the sample and the first breakpoint was detected in 10 or more background control samples; (iii) an SV with two identified breakpoints had one breakpoint present in at least 98 control samples (3% of the control samples); or (iv) both breakpoints have less than 5% read support. SVs aberrations with scores from 3 to 5 were used for all the analysis, with the exception of IG translocations.

Moreover, for the detection of the hallmark event in BL, the IG-*MYC* translocation, the DELLY[79] algorithm was used in addition. We used DELLY v0.5.9 to call simple and complex structural variants (SVs). A high confident set of somatic SVs of size >1 kb, supported by at least four read pairs, and filtered for absence in the paired normal control tissue was derived. Additionally, we removed SVs detected either in ≥1% of a set of 1105 germline samples from healthy individuals from the 1000 Genomes Project phase I or in the panel of normal samples constructed from the DELLY's consensus germline SVs called in PCAWG normal tissues sample.

**Replication timing.** Repli-Seq scores[80] were used to investigate the replication timing of mutations. Replication timing of lymphoblastoid cell lines was calculated as median Repli-Seq score of the lymphoblastoid ENCODE cell lines GM12801, GM06990, GM12812, GM12813, and GM12878.

**Detection of mutational signatures.** Unsupervised analysis of mutational signatures. An unsupervised analysis of mutational signatures based on non-negative matrix factorization (NMF) was performed on a larger cohort of 219 germinal center derived B-cell lymphomas in the framework of ICGC MMML-Seq (unpublished data). This unsupervised method has high requirements on statistical power, i.e., the number of samples and the average mutational load per sample in the input data, and application only to the BL subcohort presented in this manuscript would not have yielded stable and reliable patterns. NMF was run using the function runNmfGpu from the R software package Bratwurst (unpublished data), which provides wrapper functions for NMF solvers on graphical processing units (GPUs) using the Compute Unified Device Architecture (CUDA) 8 framework and the cudamat library. The factorization rank was varied from 2 to 15. The optimal factorization rank was obtained by simultaneously minimizing the Frobenius error, maximizing the cophenetic correlation coefficient and minimizing the Amari distance. For every factorization rank, 500 iterations over different random initializations were performed. For every initial condition, iteration over update equations was performed at most 10000 times.

Supervised analysis of mutational signatures. Supervised analysis of mutational signatures has been performed using the R package YAPSA (unpublished data). Using YAPSA, a linear combination decomposition of the mutational catalog with known and predefined signatures was computed with the function LCD complex cutoff by non-negative least squares (NNLS) using functions implemented in the R package lsei[78]. In order to increase specificity, LCD complex cutoff applies the NNLS algorithm twice: once proposing all signatures supplied by the user to the decomposition, and then after the first execution, only those signatures whose exposures, i.e., contributions in the linear combination, were higher than a certain cutoff, were kept and the NNLS was run again with the reduced set of signatures. As detectability of the different signatures may vary, the cutoffs were chosen to be signature-specific. The signature-specific cutoffs were determined in a random operator characteristic (ROC) analysis using publicly available data on mutational catalogs of 7042 cancers (507 from whole-genome sequencing and 6535 from whole exome sequencing)[81] and mutational signatures from the COSMIC database [http://cancer.sanger.ac.uk/cosmic/signatures], downloaded on January 15th, 2016. The following cut-offs were employed - AC1: 0;AC2: 0.01045942; AC3: 0.08194056; AC4: 0.01753969; AC5: 0; AC6: 0.001548535; AC7: 0.04013304; AC8: 0.242755; AC9: 0.1151714; AC10: 0.01008376; AC11: 0.09924884; AC12: 0.2106201; AC13: 0.007876626; AC14: 0.1443059; AC15: 0.03796027; AC16: 0.3674349; AC17: 0.002647962; AC18: 0.3325386; AC19: 0.1167454; AC20: 0.1235028; AC21: 0.1640255; AC22: 0.03102216; AC23: 0.03338659; AC24: 0.03240176; AC25: 0.01611908; AC26: 0.09335221; AC27: 0.009320062; AC28: 0.05616434; AC29: 0.05936213; AC30: 0.05915355.

**Detection of oncodrivers.** IntoGen (version 3.0.5)[39] was run with default settings on all somatic SNVs and indels to identify drivers amongst the 39 BL.

**Integration of different variant types.** SNVs, indels, SVs, and CNAs were integrated to account for all variant types in the recurrence analysis. Whilst all genes with SNVs or indels in coding regions (nonsynonymous, splicing, frameshift event)

and ncRNA were included, SVs and CNAs were handled differently. Any genes between the breakpoints of focal SVs (<1 Mbp) were considered. However, duplications and deletions called by SOPHIA in the range of 10 kbp and 1 Mbp had to be verified by ACEseq, discarding subclonal events with less than 0.7 copy number deviation from the average ploidy. For larger SVs only genes that were directly hit by a breakpoint were considered. Only focal CNA events (<1 Mbp) were taken into account for variant integration, as these are more likely to target specific genes within the affected region than large events such as whole chromosome arm events. To capture the precise target focal SVs and CNAs were combined and local maxima of overlapping regions with more than one event were identified.

Finally, genes affected by SNVs, indels, focal, and large SVs or genes within the CNA regions of interest were considered for the recurrence analysis and any gene affected ≥3 times was further looked up in the remaining focal CNA regions and added to the oncoprints.

**Analysis at hotspots of somatic hypermutation machinery.** Mutations in the genome region of the *MYC* gene (chr8: 128748330–128753680 bp, hg19) were analyzed for targeting the RGYW and DGYW motifs. Expected values were calculated by dividing the number of bases overlapping the motif by the number of all bases in the region and multiplying with the total number of mutations (coding and non-coding) observed and by dividing the number of G/C exchanges in the motif by the number of all G/C exchanges in the region and multiplying with the total number of mutations (non-mutations) observed at G/C positions. For statistical analysis Fisher's exact test was done.

**MYC mutation analysis at phosphorylation regions.** Mutations in exon 2 of *MYC* were analyzed for enrichment in the phosphorylation area, considering a window of $-/+$ 4 amino acids close to a phosphorylation residue. The frequency of mutations observed in phosphorylation windows was obtained by dividing the observed number of mutations in phosphorylation windows ($n = 20$) through its length ($n = 78$), while the expected frequency was determined by dividing the total number of mutations in exon 2 ($n = 31$) through its length ($n = 252$). For statistical analysis, the Fisher's exact test was used.

**Detection of germline mutations.** We employed *freebayes* (v1.1.0) (https://github.com/ekg/freebayes) in single sample- and paired-sample calling mode for discovery of single-nucleotide variants, multi nucleotide variants, and insertions/deletions <50 bp (used parameters: --min-repeat-entropy 1, --report-genotype-likelihood-max, --alternate-fraction 0.2, and --no-partial-observations). Raw variant predictions were further filtered for quality (QUAL>20, QUAL/AO>2), strand bias artifacts (SAF>1, SAR>1), read position artifacts (RPR>1, RPL>1), and normalized for consistent representation across patients with *vt* (v0.5) (https://genome.sph.umich.edu/wiki/Vt). Germline variants were annotated with the Ensembl Variant Effect Predictor (VEP) (r81, https://www.ensembl.org/info/docs/tools/vep/index.html). High impact (i.e., damaging) germline mutations were defined as frameshift, stop gain, start lost, canonical splice site, and known pathogenic non-canonical splice site variants (ClinVar; accessed 2017–02–16). Putative damaging germline mutations were removed if the estimated minor allele frequency (MAF) in at least one continental population was above 0.5%, which we judged based on 53,105 sequenced individuals that were assigned to known (control) populations and without cancer diagnosis from the ExAC resource (http://exac.broadinstitute.org), the 1000 Genomes Project (http://www.internationalgenome.org), and the NHLBI GO Exome Sequencing Project. Putative gain-of-function (GoF) missense variants in *TP53* were further evaluated based on information in the IARC *TP53* database (http://p53.iarc.fr/) and annotated as pathogenic if *TP53* mutations were classified as "non-functional" based on experimental transcriptional activity assays. Finally, all germline mutations were excluded from the analysis if annotated as benign in ClinVar.

For the detection of germline mutations in genes typically somatically mutated in the BL cohort, we additionally identified candidate germline variants using the following criteria: (i) variant has a combined annotation dependent depletion (CAAD) score higher than 13, which is considered high impact for variants; (ii) for a missense variant, gene's ExAC missense intolerance Z-score higher than 2 or, for an LoF variant, the gene's ExAC pLI score higher than 0.9; (iii) the variant was not observed in 184 non-BL gcBCL cases available locally (Huebschmann et a., under revision).

**Telomere content estimation.** The telomere content was determined from whole-genome sequencing data using the software tool TelomereHunter (www.dkfz.de/en/applied-bioinformatics/telomerehunter/telomerehunter.html) (unpublished data). In short, unmapped reads or reads with a very low alignment confidence (mapping quality lower than 8) containing six non-consecutive instances of the four most common telomeric repeat types (TTAGGG, TCAGGG, TGAGGG, and TTGGGG) were extracted. The telomere content was determined by normalizing the telomere read count to all reads in the sample with a GC-content of 48–52%. In the case of tumor samples, the telomere content was further corrected for the tumor purity (as estimated by ACEseq) using the following formula:

$T_{pure} = \frac{T - C(1-purity)}{purity}$ Where $T$ and $C$ are the telomere contents of the tumor and control sample, respectively, and $T_{pure}$ is the purity-corrected telomere content of the tumor sample.

**Transcriptome analysis**. Transcriptome data were mapped with segemehl 0.2.0,[82] allowing for spliced alignments and using a minimum accuracy of 90%. Gene expression values were counted using RNAcounter 1.5.2 [https://pypi.python.org/pypi/rnacounter], using the "--nh" option and counting only exonic reads (-t exon). Differential expression was analyzed using EdgeR with default parameters and a significance criterion of 0.05 (p-value adjusted for multiple testing)[83].

**Gene expression based BL classification**. We transferred an array-based gene expression classifier by Hummel et al.[21] to distinguish mBL and non-mBL using RNA-seq data. In short: we considered all unique genes represented by Affymetrix probe sets included in the original classifier that were available in the RNA-seq data (37 upregulated in BL; 14 downregulated in BL). An initial set of 48 solid lymphoma samples was classified based on histopathology and FISH into BL and DLBCL. For these cases, RNA-seq data for the exemplarily selected regions was analyzed on the average reads per million (RPM) scale. Expression values were log transformed and standardized. Unsupervised hierarchical clustering resulted in a perfect separation of histopathologically defined BL and DLBCL as well as up- and downregulated probe sets. We utilized the classification based on histopathology to estimate the density of expression values separately for BL and DLBCL samples assuming normal distribution of expression values for each probe set in each subgroup. For classification, the conditional probability for BL is estimated based on the observed expression and the estimated subgroup specific distributions. We applied the median of the estimated BL probabilities over all exemplarily selected regions as an overall classification score for BL vs DLBCL. Given the results of the testset with 48 histopathologically classified lymphomas we applied the thresholds of <0.25 for non-mBL and ≥0.66 for mBL. Despite being developed for solid BL, we also applied the RNA-seq based classifier to leukBL and compared to 13 prototypic non-BL cases (DLBCL) from the ICGC MMML-seq cohort (Supplementary Fig. 1b).

**Fusion transcripts analysis**. For all the fusion genes detected by WGS, we checked the presence of fusion transcripts by usage of two different tools, segemehl and confuse. To define the presence of fusion transcripts the junction of the fusion transcripts detected by both tools were used. Segemehl is a splice-aware mapper that can detect reads with splice sites independent of annotation. Segemehl alignment files were searched for reads that originated in the vicinity of one of the genes containing the break point and were spliced to the vicinity of the other gene, with a maximum distance to the gene of 20k bases. If a sufficient amount of spliced reads (above 3) were found, all other ICGC MMML-Seq RNA-seq alignment files were checked for existence of the respective splice site. Only splice sites that were unique to the dataset containing the break point were reported.

ConFuse is a novel downstream filtering tool for reliably selecting high-confidence fusion candidates. It takes multiples features into account to assign each fusion transcript a confidence score[84]. These features are mainly related to mapping artifact, mapping quality, number of supporting reads (spanning reads and split reads), and structural motifs. ConFuse classifies the fusion transcripts into three categories (high-, medium- and low-confidence) based on the confidence score. It can prioritize the fusion candidates for further analysis and experimental validation.

**Differential alternative splicing analysis**. Differential splicing analysis was conducted using DIEGO[85], with default parameters and a significance criterion of 0.05 (adjusted p-value).

**PCR and Sanger sequencing**. Selected SNVs and SVs were amplified by polymerase chain reaction from the genomic DNA using specific primers. Amplicons were purified (MinElute 96 UF PCR Purification Kit, Qiagen, Hilden, Germany) and cycle-sequenced using fluorescent dye-termination (Big Dye Terminator V1.1 Cycle Sequencing Kit, Applied Biosystems, Darmstadt, Germany) and an ABI 3100 or ABI 310 automatic capillary genetic analyzer.

In addition, a subset of fusion transcripts were validated using specific primers to amplify breakpoint fusion sequence. The RNAs from the tumor samples were treated with DNase I (RNAase-Free, Ambion, Thermofisher Scientific, Darmstadt, Germany). Complementary DNA (cDNA) was synthesized from 1 μg of total RNA using Quanti Tect Reverse Transcription kit (Qiagene, Hilden, Germany) according to manufacturer´s instruction. The cDNAs were amplified by polymerase chain reaction using specific primers and conditions and the amplicons were purified and sequenced according to the conditions described above.

**TCF3-ID3-CBFA2T3 interaction modeling**. We modeled the TCF3-ID3 dimer using Modeller[86] and the ternary complex structure between the *Tcf3-Neurod1* dimer and DNA (PDB ID: 2QL2) as a template. The TCF3-CBFA2T3 interaction interface was predicted using Interprets[87] and the structure between *TCF12-RUNX1T1* (PDB ID: 2KNH) as a template. Electrostatic surfaces for the ID3-TCF3 dimers (considering both E12 and E47 isoforms) were calculated through the APBS approach available in Pymol.

**Reporting summary**. Further information on experimental design is available in the Nature Research Reporting Summary linked to this article.

## Data availability

WGS and RNA-Seq alignments are available from the European Genome-phenome archive (EGA) under the accession numbers: EGA-S00001002198 & EGAS00001001692.

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

## Acknowledgements

This study has been supported by the German Ministry of Science and Education (BMBF) in the framework of the ICGC MMML-Seq (01KU1002A-J), ICGC DE-Mining (01KU1505G and 01KU1505E), and MMML-MYC-SYS (036166B) projects and the KinderKrebsInitiative Buchholz Holm-Seppensen. This work was also supported by the BMBF-funded Heidelberg Center for Human Bioinformatics (HD-HuB) within the German Network for Bioinformatics Infrastructure (de.NBI) (#031A537A, #031A537C). We thank the High Throughput Sequencing unit of the DKFZ Genomics and Proteomics Core Facility for providing whole-genome sequencing services. Former grant support of MMML by the Deutsche Krebshilfe (2003–2011) is gratefully acknowledged. C.L. was supported by an Alexander von Humboldt Foundation post-doctoral fellowship. S.M.W. received funding through a SNSF Early Postdoc Mobility fellowship (P2ELP3_155365) and an EMBO Lon-Term Fellowship (ALTF 755–2014). The support of the technical staff of the Institutes of Human Genetics in Kiel and Ulm is gratefully acknowledged. The authors acknowledge the ICGC MMML- Seq and ICGC DE-Mining Projects on whose behalf this work was carried out.

## Author contributions

A.K., A.C., A.R., A.K.B., L.T., ML.H., P.M., S.S., M.R., H.S., and B.Burkhardt provided tumor samples and clinical data. M.A.W. and R.K. provided normal B-cell samples. P.M. coordinated and performed pathology review. D.L. and M.S. stained and reviewed cryomaterial. M.H. and W.K. prepared and performed analyte quality control. B.Brors. and M.B.S performed next-generation sequencing analyses. M.L., M.Schlesner, M.Z., P.L., P.R., R.E., S.H. B.Brors., and J.O.K supervised next-generation sequencing analysis and interpreted data. C.Lawerenz performed and coordinated data transfer and data management of NGS data. B.R., D.H., J.H., K.K., M.K., N.P., S.Sungalee, S.M.W., U.H.T., Z. H., L.S., J.S., L.F. and M.Schlesner performed analysis of next-generation sequencing data. A.H., J.R., O.A., R.W., S.M.A., J.B., and C.L. performed validation analyses. C.L. and R.S. performed FISH validation analyses. R.S., M.K., and M.L. provided and analyzed data from the MMML cohort. C.L., D.H., K.K., R.K., R.S., SH.B., S.M.A., UH.T., M.R., M. Schlesner, R.W., and B.Burkhardt interpreted data and wrote the manuscript. R.K., R.S., B.Burkhardt, S.H., and M.Schlesner designed the study. A.H., A.K.B., J.R., R.W, SM.A., and C.L. supported coordination of the project. R.S. coordinated the ICGC MMML-Seq network. All authors read and approved the final manuscript.

## Additional information

**Competing interests:** The authors declare no competing interests.

Cristina López [1,2], Kortine Kleinheinz [3,4], Sietse M. Aukema[2,5], Marius Rohde[6], Stephan H. Bernhart[7,8,9], Daniel Hübschmann[3,10,11], Rabea Wagener[1,2], Umut H. Toprak[3,12,13], Francesco Raimondi[14], Markus Kreuz[15], Sebastian M. Waszak [16], Zhiqin Huang[17], Lina Sieverling[13,18], Nagarajan Paramasivam[3,19], Julian Seufert[12], Stephanie Sungalee [16], Robert B. Russell [14], Julia Bausinger[1], Helene Kretzmer[7,8,9,20], Ole Ammerpohl[1], Anke K. Bergmann[2,21], Hans Binder[7,8], Arndt Borkhardt [22], Benedikt Brors [18], Alexander Claviez[21], Gero Doose[7,8,9], Lars Feuerbach[18], Andrea Haake[2], Martin-Leo Hansmann[23], Jessica Hoell[22], Michael Hummel[24], Jan O. Korbel [16], Chris Lawerenz[3], Dido Lenze[24], Bernhard Radlwimmer[17], Julia Richter [2,5], Philip Rosenstiel[25], Andreas Rosenwald[26], Markus B. Schilhabel[25], Harald Stein[27], Stephan Stilgenbauer[28], Peter F. Stadler [8], Monika Szczepanowski[5], Marc A. Weniger[29], Marc Zapatka [17], Roland Eils [3,4], Peter Lichter[17], Markus Loeffler[15], Peter Möller[30], Lorenz Trümper[31], Wolfram Klapper[5], ICGC MMML-Seq Consortium, Steve Hoffmann[7,8,9,32], Ralf Küppers [29], Birgit Burkhardt[33], Matthias Schlesner [3,12] & Reiner Siebert[1,2]

[1]Institute of Human Genetics, Ulm University and Ulm University Medical Center, 89081 Ulm, Germany. [2]Institute of Human Genetics, Christian-Albrechts-University, 24105 Kiel, Germany. [3]Division of Theoretical Bioinformatics (B080), German Cancer Research Center (DKFZ), 69120 Heidelberg, Germany. [4]Department for Bioinformatics and Functional Genomics, Institute of Pharmacy and Molecular Biotechnology and Bioquant,

University of Heidelberg, 69120 Heidelberg, Germany. [5]Hematopathology Section, Christian-Albrechts-University, 24105 Kiel, Germany. [6]Pediatric Hematology and Oncology, University Hospital Giessen, 35392 Giessen, Germany. [7]Interdisciplinary Center for Bioinformatics, University of Leipzig, 04107 Leipzig, Germany. [8]Bioinformatics Group, Department of Computer, University of Leipzig, 04107 Leipzig, Germany. [9]Transcriptome Bioinformatics, LIFE Research Center for Civilization Diseases, University of Leipzig, 04107 Leipzig, Germany. [10]Department of Pediatric Immunology, Hematology and Oncology, University Hospital, 69120 Heidelberg, Germany. [11]German Cancer Research Center (DKFZ), Division of Stem Cells and Cancer, Heidelberg, Germany and Heidelberg Institute for Stem Cell Technology and Experimental Medicine (HI-STEM gGmbH), 69120 Heidelberg, Germany. [12]Bioinformatics and Omics Data Analytics (B240), German Cancer Research Center (DKFZ), 69120 Heidelberg, Germany. [13]Faculty of Biosciences, Heidelberg University, 69120 Heidelberg, Germany. [14]Cell Networks, Bioquant and Biochemistry CenterBiochemie Zentrum Heidelberg (BZH), University of Heidelberg, 69120 Heidelberg, Germany. [15]Institute for Medical Informatics Statistics and Epidemiology, 04107 Leipzig, Germany. [16]Genome Biology Unit, EMBL Heidelberg, 69117 Heidelberg, Germany. [17]Division of Molecular Genetics, German Cancer Research Center (DKFZ), 69120 Heidelberg, Germany. [18]Division of Applied Bioinformatics (G200), German Cancer Research Center (DKFZ), 69120 Heidelberg, Germany. [19]Medical Faculty Heidelberg, Heidelberg University, 69120 Heidelber, Germany. [20]Department of Genome Regulation, Max Planck Institute for Molecular Genetics, 14195 Berlin, Germany. [21]Department of Pediatrics, University Hospital Schleswig-Holstein, Campus Kiel, 24105 Kiel, Germany. [22]Medical Faculty, Department of Pediatric Oncology, Hematology and Clinical Immunology, Heinrich-Heine-University, 40225 Düsseldorf, Germany. [23]Senckenberg Institute of Pathology, University of Frankfurt Medical School, 60590 Frankfurt am Main, Germany. [24]Institute of Pathology, Charité – University Medicine Berlin, 10117 Berlin, Germany. [25]Institute of Clinical Molecular Biology, Christian-Albrechts-University, 24105 Kiel, Germany. [26]Institute of Pathology, Comprehensive Cancer Center Mainfranken, University of Würzburg, 97080 Würzburg, Germany. [27]Pathodiagnostik Berlin, 12099 Berlin, Germany. [28]Department for Internal Medicine III, Ulm University, 89081 Ulm, Germany. [29]Institute of Cell Biology (Cancer Research), Medical School, University of Duisburg-Essen, 45147 Essen, Germany. [30]Institute of Pathology, University of Ulm and University Hospital of Ulm, 89081 Ulm, Germany. [31]Department of Hematology and Oncology, Georg-August-University of Göttingen, 37075 Göttingen, Germany. [32]Computational Biology, Leibniz Institute on Ageing-Fritz Lipmann Institut (FLI), 07745 Jena, Germany. [33]University Hospital Münster - Pediatric Hematology and Oncology, 48149 Münster, Germany. The authors contributed equally: Cristina López, Kortine Kleinheinz, Sietse M. Aukema, Marius Rohde, Stephan H. Bernhart, Daniel Hübschmann. These authors jointly supervised this work: Steve Hoffmann, Ralf Küppers, Birgit Burkhardt, Matthias Schlesner, Reiner Siebert.

## ICGC MMML-Seq Consortium

Susanne Wagner[2], Gesine Richter[2], Jürgen Eils[3], Jules Kerssemakers[3], Christina Jaeger-Schmidt[3], Ingrid Scholz[3], Christoph Borst[34], Friederike Braulke[31], Martin Dreyling[35], Sonja Eberth[35], Hermann Einsele[36], Norbert Frickhofen[37], Siegfried Haas[34], Dennis Karsch[38], Nicole Klepl[31], Michael Kneba[38], Jasmin Lisfeld[33], Luisa Mantovani-Löffler[39], German Ott[40], Christina Stadler[31], Peter Staib[41], Thorsten Zenz[42], Dieter Kube[31], Ulrike Kostezka[43], Vera Binder[22], Ellen Leich[26], Inga Nagel[2], Jordan Pischimariov[26], Stefan Schreiber[44], Inga Vater[2], Lydia Hopp[7], David Langenberger[7,8,9] & Maciej Rosolowski[15]

[34]Department of Internal Medicine/Hematology, Friedrich-Ebert-Hospital, Neumünster 24534, Germany. [35]Department of Medicine III - Campus Grosshadern, University Hospital Munich, Munich 81377, Germany. [36]University Hospital Würzburg, Department of Medicine and Poliklinik II, University of Würzburg, Würzburg 97080, Germany. [37]Department of Medicine III, Hematology and Oncology, Dr. Horst-Schmidt-Kliniken of Wiesbaden, Wiesbaden 65199, Germany. [38]Department of Internal Medicine II: Hematology and Oncology, University Medical Centre, Campus Kiel, Kiel 24105, Germany. [39]Hospital of Internal Medicine II, Hematology and Oncology, St-Georg Hospital Leipzig, Leipzig 04129, Germany. [40]Department of Pathology, Robert-Bosch-Hospital, Stuttgart 89081, Germany. [41]Clinic for Hematology and Oncology, St.-Antonius-Hospital, Eschweiler 52249, Germany. [42]National Centre for Tumor Disease, Heidelberg 69120, Germany. [43]Comprehensive Cancer Center Ulm (CCCU), University Hospital Ulm, Ulm 89081, Germany. [44]Department of General Internal Medicine, University Kiel, Kiel 24105, Germany

