## [Peer Review File · Nature Communications]

Reviewers' comments:

Reviewer #1 (Remarks to the Author):

In this manuscript, Lopez et al describe the comprehensive genomic and transcriptomic characterization of sporadic (non-EBV) Burkitt lymphoma using a combination of whole-genome and transcriptome sequencing. To my knowledge this is the largest cohort to date of BL cases investigated by WGS. The authors include an impressive array of analysis including Ig-MYC breakpoint analysis, somatic SNVs at the MYC locus, and non-MYC cooperating events (other translocations, driver mutations, differential splicing, and germline mutations).

In particular, the authors highlight (1) an enrichment of IGHA-MYC rearrangements in sBL when compared to other B cell lymphoma types, (2) variable transcripts derived from the MYC locus including several antisense IGH-MYC fusions, (3) somatic SNVs clustered around MYC PTM sites, (4) novel non-Ig-MYC rearrangements leading to deregulation of putative oncogenes through IGK-CCNG1 and IGH-HECW2, (5) detectable gene fusions including 8 fusion transcripts, (6) somatic alterations cancer driver genes and non-coding RNAs, (7) differentially spliced driver genes including TCF3, (8) germline events in known cancer predisposition genes such as FANCG and also in established BL driver genes, and (9) mutational signatures are in large part driven by lymphocyte processes (class switch recombination, somatic hypermutation).

Overall this study represents a comprehensive genomic survey of this subtype of BL and will be a valuable resource for the lymphoma genomics community. The manuscript is clearly written, and the computational methods clearly described. Although many of the events detected in this study have been described previously, there are several genomic and transcriptomic events discovered here that are predicted to be novel drivers of BL (CCNG1/HECW2 deregulation, new fusion transcripts, upregulation of TCF3 E47); however, these events are not followed up with any mechanistic functional experiments to establish their relevance.

A few additional questions I have of the results-

1) The RFX7 mutations are strongly mutually exclusive of CCND3 alterations - are these factors known to be functioning in the same pathway? Can the authors speculate as to how they may be related?

2) Is the differential expression of E47 and E12 TCF3 isoforms observed in another other tumor type (or other lymphoma types), or in stem cells? Or is this expression pattern unique to BL?

Reviewer #2 (Remarks to the Author):

Lopez and colleagues have applied whole genome sequencing and RNA-seq to primary tumour samples from 39 cases of sporadic Burkitt lymphoma. Through the subsequent application of state-of-the-art bioinformatic analyses, they shed considerable light on the mutational and transcriptional landscape of this relatively uncommon but nonetheless important neoplasm.

The manuscript is exceptionally well-written and the presentation of the findings is admirably precise. The manuscript is lengthy but this seems justified by the abundance of the interesting findings.

This is an exploratory exercise; functional implications are pointed out thoughtfully but not pursued experimentally. The work benefits from the exceptional access that these authors possess to samples and technology. Some findings are novel while many others represent valuable confirmation of previously published findings.

I really have no serious concerns about this work but offer the following minor suggestions.

1. As an aggressive neoplasm of mature B-lymphoid cells Burkitt lymphoma is closely related to diffuse large B-cell lymphoma, especially the "GCB" subtype. The authors might consider developing further the implications of their findings for understanding the distinctive pathogenic mechanisms that might be involved in these two neoplasms as well as the implications for their pathological diagnosis and clinical management.
2. Recent work has illuminated the mutational and transcriptomic features of endemic Burkitt lymphoma. Perhaps the current work might address some of the differences between endemic and sporadic Burkitt lymphoma.
3. Figure 1f might be improved by more detailed annotation of the functional domains of MYC. In particular, the domains MBI and MBII should be shown.

Reviewer #1:

In this manuscript, Lopez et al describe the comprehensive genomic and transcriptomic characterization of sporadic (non-EBV) Burkitt lymphoma using a combination of whole-genome and transcriptome sequencing. To my knowledge this is the largest cohort to date of BL cases investigated by WGS. The authors include an impressive array of analysis including Ig-MYC breakpoint analysis, somatic SNVs at the MYC locus, and non-MYC cooperating events (other translocations, driver mutations, differential splicing, and germline mutations).

In particular, the authors highlight (1) an enrichment of IGHA-MYC rearrangements in sBL when compared to other B cell lymphoma types, (2) variable transcripts derived from the MYC locus including several antisense IGH-MYC fusions, (3) somatic SNVs clustered around MYC PTM sites, (4) novel non-Ig-MYC rearrangements leading to deregulation of putative oncogenes through IGK-CCNG1 and IGH-HECW2, (5) detectable gene fusions including 8 fusion transcripts, (6) somatic alterations cancer driver genes and non-coding RNAs, (7) differentially spliced driver genes including TCF3, (8) germline events in known cancer predisposition genes such as FANCG and also in established BL driver genes, and (9) mutational signatures are in large part driven by lymphocyte processes (class switch recombination, somatic hypermutation).

Overall this study represents a comprehensive genomic survey of this subtype of BL and will be a valuable resource for the lymphoma genomics community. The manuscript is clearly written, and the computational methods clearly described. Although many of the events detected in this study have been described previously, there are several genomic and transcriptomic events discovered here that are predicted to be novel drivers of BL (CCNG1/HECW2 deregulation, new fusion transcripts, upregulation of TCF3 E47); however, these events are not followed up with any mechanistic functional experiments to establish their relevance.

We thank the reviewer for these reassuring comments. We appreciate that the reviewer highlights our comprehensive analyses, in which we have shown that diverse molecular mechanisms including enhancer hijacking, gene fusion, alternative splicing, as well as germline and somatic mutations can complement each other in the dysregulation of driver genes and pathways of BL. In the present study we were interested in the mechanistic and functional ramifications caused by “experiments of nature” with regard to deregulation of genes involved in the pathogenesis of BL. While functional experiments in artificial test systems were out of scope of this study, a wealth of *in vitro* data does exist for many of these genes (e.g. *MYC*, *TCF3* or *ID3*).

A few additional questions I have of the results-

- 1) The RFX7 mutations are strongly mutually exclusive of CCND3 alterations - are these factors known to be functioning in the same pathway? Can the authors speculate as to how they may be related?

The reviewer raises an interesting point. To address this, we performed a series of bioinformatics analyses and explored databases and as well as scientific literature to investigate the potential relation between these two genes/proteins and whether they are involved in the same pathway. There is indeed

some limited evidence of a functional relationship between these two genes. Though they have not yet been annotated in any single tight biological process or pathway, there are hints from several sources that they are both involved in regulating the cell cycle. We added the results of these additional studies to the revised manuscript on page 14 as follows: "Mutual exclusivity argues that there could be a functional connection between RFX7 and CCND3. Although there is no pathway or discrete gene ontology group containing both proteins, there is evidence that both share upstream regulatory genes (Marbach et al., 2016) and that they participate in the same tissue-specific functional interaction network (<http://hb.flatironinstitute.org/gene/896+64864>). Moreover, RFX7 (and other RFX transcription factors) is co-expressed with CCNK, a regulatory subunit of cyclin dependent kinases (Aftab et al., 2008) and RFX7 is described as transcription factor for the oncogene *MYC*. Taken together, these findings suggest that RFX7 could be a transcription factor regulating genes in cell cycle control (including *CCND3*). The precise role of this putative cell-cycle related process in BL will require additional investigation."

2) Is the differential expression of E47 and E12 TCF3 isoforms observed in another other tumor type (or other lymphoma types), or in stem cells? Or is this expression pattern unique to BL?

The reviewer proposes additional analyses in other tumors types (particularly lymphoma) and stem cells to further investigate the observed differential expression of E47 and E12 TCF3 isoforms in BL. Accordingly, we have analyzed RNA-seq data from other types of mature germinal center derived B-cell lymphomas generated within the ICGC MMML-Seq consortium and being part of a manuscript currently under revision in Nature Communications (NCOMMS-18-18634-T, quoted below as unpublished). We have added the results to the revised manuscript on page 15 as follows: "We extended the analysis of the differential expression of the E47 and E12 TCF3 isoforms to adult GCB-lymphomas other than BL. To this end, we mined RNA-seq data from FL (n=85), DLBCL (n=72), and FL-DLBCL (n=17) generated within the ICGC MMML-Seq consortium using the same sequencing pipeline (unpublished data). In this non-BL cohort the preferential usage of the isoform E47 is significantly lower than in BL (BL vs non-BL, $p < 0.001$) (Supplementary Fig. 8). Given that a recent study has identified differential expression of the E12 and E47 TCF3 isoforms in pluripotent human embryonic stem cells (ESCs) compared to differentiated cells we also extended our analysis to published data from ESCs and induced pluripotent stem cells (iPSCs) (Friedi et al., 2014, Yamazaki et al., 2018) (Supplementary Fig. 8). Indeed, we corroborated the relatively high expression of E12 as compared to a relatively low expression of E47 in ESCs and iPSCs cells (joined ESCs/iPSCs vs GCB cells $p = 0.002$). Thus, differential expression of the E47 and E12 TCF3 isoforms seems to be a common means to regulate TCF3 function. Skewing towards the E47 expression seems to be

particularly pronounced in BL as compared to other mature GCB-lymphomas and, thus, likely contributes to deregulation of the TCF3/ID3 complex particularly in BL lacking *ID3* and/or *TCF3* mutations.”

We display the results of our additional analyses in a new Supplementary Fig. 8. and in its legend:

Supplementary Figure 8. TCF3 splicing

Boxplot showing the ratio of spliced reads that are directly spliced to exon E47 vs spliced reads that are spliced to exon E12. While half of the spliced reads splice to E47 in GCB cells (green), most of the BL cases have roughly two thirds of the spliced reads splicing to E47 (grey boxes) (all BL vs GCB cells, $p < 0.001$; see Figure 5b for comparison of ID mutated and unmutated BL). A higher fraction of reads excluding E12 than in GCB cells (green) is also observed, though only as a trend, in the non-BL group ($p = 0.052$), reaching significance in FL ($n = 85$, light blue box, $p = 0.008$) and FL-DLBCL ($n = 17$, bluish-grey box, $p = 0.031$), whereas no significant difference to GCB cells is present in DLBCL ($n = 72$, dark blue dark box, $p = 0.444$). In contrast, in the ESC ($n = 4$, orange) and IPSC ($n = 6$, yellow) the splice reads predominately splice to the E12 exon ($p = 0.024$ and $p = 0.008$, respectively, for comparison to GCB cells). For analyses of DLBCL, FL and FL-DLBCL, in which *ID3* mutations are rare (see Supplementary Figure

9), RNA-seq data of the ICGC MMML-Seq consortium (unpublished), which have been generated using the same sequencing pipeline like the one here applied to BL, were mined. For the analyses of ESC and iPSC we mined data from Friedi et al., 2014 and used the alignment tool STAR for mapping splice read. For statistical analysis the Wilcoxon rank sum test was applied and p-values were adjusted for multiple testing according to Benjamini-Hochberg.

Reviewer #2:

Lopez and colleagues have applied whole genome sequencing and RNA-seq to primary tumour samples from 39 cases of sporadic Burkitt lymphoma. Through the subsequent application of state-of-the-art bioinformatic analyses, they shed considerable light on the mutational and transcriptional landscape of this relatively uncommon but nonetheless important neoplasm.

The manuscript is exceptionally well-written and the presentation of the findings is admirably precise. The manuscript is lengthy but this seems justified by the abundance of the interesting findings.

This is an exploratory exercise; functional implications are pointed out thoughtfully but not pursued experimentally. The work benefits from the exceptional access that these authors possess to samples and technology. Some findings are novel while many others represent valuable confirmation of previously published findings.

We thank the reviewer for the positive evaluation of the submitted manuscript and are grateful that he/she values that we have contributed with our findings, some of them novel and other previously described, to a more comprehensive understanding of Burkitt lymphoma. We are pleased that the reviewer justified the length of the manuscript which is due to the amount of data included. With regard to functional implications we would like to point out (as already indicated to reviewer 1) that in the present study we were interested in the mechanistic and functional ramifications caused by “experiments of nature” with regard to deregulation of genes involved in the pathogenesis of BL. While functional experiments in artificial test systems were out of scope of this study, a wealth of *in vitro* data does exist for many of these genes (e.g. *MYC*, *TCF3* or *ID3*).

I really have no serious concerns about this work but offer the following minor suggestions.

- 1) As an aggressive neoplasm of mature B-lymphoid cells Burkitt lymphoma is closely related to diffuse large B-cell lymphoma, especially the "GCB" subtype. The authors might consider developing further the implications of their findings for understanding the distinctive pathogenic mechanisms that might be involved in these two neoplasms as well as the implications for their pathological diagnosis and clinical management.

We thank the reviewer for this comment. Obviously, we fully agree that Burkitt lymphoma belongs to the group of germinal center derived B-cell lymphomas. The problem we see with the

proposed comparison of BL to diffuse large B-cell lymphoma (DLBCL) is that the latter predominantly arises in patients of the 6th decade whereas BL arise predominantly in the pediatric age group. Hence, there might be a strong age bias in such an analysis. Indeed, we and others have recently shown that molecular features of DLBCL significantly change with age (see, e.g., Klapper et al., Blood, 2012; Paul et al., Leuk Lymph, 2018). Moreover, pediatric BL and adult DLBCL are (at least in Germany) treated according to different treatment protocols, i.e. they are managed differently in the clinics. Despite these aspects, we agree with the reviewer that it might be interesting to compare the mutational profile of the pediatric BL studied herein with DLBCL. In addition to the full DLBCL population, we also analyzed a subset of pediatric DLBCL showing an age distribution similar to the cohort of BL analyzed by us. Accordingly, we mined the mutational data of recently published cohorts of DLBCL (including studies by Morin et al., 2013; Reddy et al., 2017; Arthur et al., 2018; Chapuy et al., 2018; Schmitz et al., 2018) and our own analyses (Huebschmann et al., NCOMMS-18-18634-T) for pediatric patients. Unfortunately, the cohorts either did not contain pediatric patients or the data were not readily available in the public domain. Fortunately, the study by Reddy et al. (2017) of 1001 DLBCL includes 10 pediatric DLBCL (<18 years at diagnosis). Unfortunately Reddy et al. published solely the mutation status of the 150 driver genes identified in the study for every case. Of the 20 genes frequently mutated in the present series of sBL ($\geq 15\%$ of sBL cases) a total of 10 were not delineated as drivers in DLBCL by Reddy et al. The results are summarized in a **new Supplementary Fig. 9a and in its legend (see below)**.

The GCB-type lymphomas do not only encompass DLBCL but also follicular lymphoma (FL). Thus, to answer the comment of the reviewer accurately, we compared the mutational landscape of sBL not only with pediatric DLBCL but also with pediatric type follicular lymphoma (PTFL). For those, either no complete exome data (Ozawa et al., 2016) or only limited (12-14 genes) sequencing data linked to age at diagnosis (Schmidt et al., 2016; Schmidt et al., 2018) are publically available. Thus, we decided to focus this comparison on those genes reported by Louissaint A Jr et al., 2016 to be recurrently mutated in PTFL and affected in patients <18 years (Figure 2A in Louissaint A Jr et al., 2016). This comparison shows, that those 5/6 genes mutated in PTFL (<18 yrs) do not show protein-changing SNVs in sBL including *TNFRSF14* and *MAP2K1*. The exception to this was *ARID1A*, which was mutated in both entities but at higher frequency in sBL than PTFL. The results of these comparisons are summarized in a **new Supplementary Fig. 9b as well as its legend which is depicted below**.

Supplementary Figure 9. Comparison of the mutational landscape of pediatric sBL vs pediatric diffuse large B-cell lymphoma (DLBCL) and pediatric-type follicular lymphoma (PTFL)

a Comparison of recurrently mutated genes in pediatric sBL (n=20) to driver genes (n=150) identified in an overall cohort of 1001 DLBCL by Reddy et al., 2017. This cohort contains a subset of 10 pediatric DLBCL (≤ 18 years at diagnosis). On the left side, the Venn diagram shows that only 10 of the 20 recurrently mutated genes in sBL are reported as driver genes in DLBCL. Of those 150 driver genes, only 14 are recurrently mutated in pediatric DLBCL, i.e. affected in $\geq 2/10$ pediatric DLBCL cases. The diagram on the right side shows the relative amount of sBL and pediatric DLBCL cases being mutated in those 14 genes. Eight of these DLBCL driver genes recurrently mutated in pediatric DLBCL are not at all mutated in our sBL cohort and four in less than 10% of the sBL cases.

b Comparison of the mutational landscape of pediatric sBL to mutations in 17 genes identified by Louissaint A Jr et al. as frequently mutated in 24 PTFL (according to Figure 2A in Louissaint A Jr et al., 2016). This cohort contains a subset of 14 PTFL ≤ 18 years at diagnosis. On the left side, the Venn diagram shows that only 3 of the 20 frequently mutated genes in sBL are recurrently mutated in PTFL. Of the 17 frequently mutated genes in PTFL, only 6 are mutated in PTFL ≤ 18 years at diagnosis, i.e. affected in at least $1/14$ PTFL ≤ 18 years at diagnosis. The diagram on the right side shows the relative amount of sBL

and PTF1 < 18 years at diagnosis being mutated in those 6 genes. Five of these 6 genes are not at all mutated in our sBL cohort.

Finally, to point the reader to these additional analyses, we have included into the revised manuscript on page 18 the following sentence: "The overlap of affected genes in pediatric sBL on the one hand and in other GCB-lymphomas occurring in children like pediatric DLBCL (Reddy et al., 2017) or pediatric-type FL (Louissaint A Jr et al., 2016) on the other hand is small (Supplementary Fig. 9)."

2) Recent work has illuminated the mutational and transcriptomic features of endemic Burkitt lymphoma. Perhaps the current work might address some of the differences between endemic and sporadic Burkitt lymphoma.

We thank the reviewer for the interesting comment. The scope of this study was the analysis of the genomic and transcriptomic alterations in sporadic Burkitt lymphoma. Accordingly, we do not have own data on genomic and transcriptomic analysis of endemic BL at hand with the exception of few cell lines. Nevertheless, we used the published datasets on sequencing analyses of eBL from Abate et al., 2015 (including 20 eBL cases) and Kaymaz et al., 2017 (including 28 eBL cases) to compare the mutational profile of eBL to our series of sBL. As already published in the respective studies by Abate et al. 2015 and Kaymaz et al., 2017, sBL and eBL share a comparable mutational landscape with regard to the genes affected, but the mutation frequencies differ: eBL harbor a lower frequency of mutations in genes like *MYC*, *ID3*, *CCND3* or *SMARCA4*. In contrast, genes like *DDX3X*, *TCF3* or *GNAI2* are mutated in a higher proportion of eBL cases than sBL. Furthermore, some genes like *CCNF*, *PRRC2C*, *BCL7A*, *PRKDC* and *RPRD2* have been reported to be mutated solely in eBL. These new analyses and results are summarized in a new Supplementary Fig. 10 and in its legend:

Supplementary Figure 10. Comparison of pediatric sBL vs eBL

a/b Comparison of the frequently mutated genes in sBL (grey) and eBL based on analyses of eBL published by Kaymaz et al., 2017 (a, orange) as well as Abate et al., 2015 (b, green). Only genes which were mutated in more than 10% of sBL (left sides) or in more than 10% of eBL (right side) of the respective studies were included. The comparison shows that overall sBL and eBL share a comparable mutational landscape with regard to affected genes, but with some variation in the percentages of cases affected per gene. Whereas eBL harbor a lower amount of cases with mutations in genes like *MYC*, *ID3*, *CCND3* or *SMARCA4*, they show a higher frequency of mutations in genes like *DDX3X*, *TCF3* or *GNAI2*. These findings are in line with the observations published by Kaymaz et al., and Abate et al.,.

In addition, in the revised manuscript in page “18”, we have added the following sentence to introduce the comparisons on the mutational landscapes between sBL and eBL performed above: “In contrast, there is a considerable overlap with regard to genes affected in eBL despite some differences in mutation frequencies (Abate et al. 2015 and Kaymaz et al., 2017) (Supplementary Fig. 10)”.

Finally, with regard to transcriptomic data this was hampered by the lack of RNA-seq data of eBL within our cohort. We nevertheless used the list of genes reported to be differentially expressed in eBL in comparison to sBL recently published in Kaymaz et al., 2017. We analyzed the expression of these genes in our sBL and normal germinal center B-cells samples. This explorative analysis revealed an inconclusive pattern of up- and downregulated genes. Thus, we refrained from including any analyses of transcriptome comparisons due to the lack of appropriate datasets and materials to generate them (i.e. fresh materials from primary eBL).

3) Figure 1f might be improved by more detailed annotation of the functional domains of MYC. In particular, the domains MBI and MBII should be shown.

We thank the reviewer to make us aware on this point. In the revised manuscript, we modified Figure 1f, in accordance with the reviewer’s suggestions. In particular, we added the information of the localization of the functional domains of the MYC protein including the Myc box I (MBI) and Myc box II (MBII) to make this figure more comprehensive.

New Figure 1f

REVIEWERS' COMMENTS:

Reviewer #1 (Remarks to the Author):

The authors have satisfactorily addressed all of my comments.

Reviewer #2 (Remarks to the Author):

In my judgment, the authors have been meticulous in addressing all of the points raised in the reviews.